# Measurement of radioactivity in soils of Karamjal and Harbaria mangrove forest of Sundarbans for establishment of radiological database

**M. M. Mahfuz Siraz** [1]*, **Jubair A. M.**[2], **M. S. Alam**[2], **Md. Bazlar Rashid**[3], **Z. Hossain**[1], **Mayeen Uddin Khandaker**[4,5], **D. A. Bradley**[5,6], **S. Yeasmin**[1]

**1** Health Physics Division, Atomic Energy Centre, Dhaka, Bangladesh, **2** Department of Nuclear Engineering, University of Dhaka, Dhaka, Bangladesh, **3** Geological Survey of Bangladesh, Segunbaghicha, Dhaka, Bangladesh, **4** Department of General Educational Development, Faculty of Science and Information Technology, Daffodil International University, Dhaka, Bangladesh, **5** Centre for Applied Physics and Radiation Technologies, School of Engineering and Technology, Sunway University, Bandar Sunway, Petaling Jaya, Selangor, Malaysia, **6** Centre for Nuclear and Radiation Physics, Department of Physics, University of Surrey, Guildford, Surrey, United Kingdom

* mahfuzsiraz1985@yahoo.com

**Data Availability Statement:** All relevant data are within the paper.

## Abstract

This work presents the first in-depth study of soil radioactivity in the mangrove forest of Bangladesh part of the Sundarbans. It used HPGe gamma-ray spectrometry to measure the amount of natural radioactivity in soil samples from Karamjal and Harbaria sites of the world's largest mangrove forest. The activity concentrations of most of the $^{226}$Ra (14±2 Bqkg$^{-1}$ to 35±4 Bqkg$^{-1}$) and $^{232}$Th (30±5 Bqkg$^{-1}$ to 50±9 Bqkg$^{-1}$) lie within the world average values, but the $^{40}$K concentration (370± 44 Bqkg$^{-1}$ to 660±72 Bqkg$^{-1}$) was found to have exceeded the world average value. The evaluation of radiological hazard parameters revealed that the outdoor absorbed dose rate (maximum 73.25 nGyh$^{-1}$) and outdoor annual effective dose (maximum 0.09 mSvy$^{-1}$) for most samples exceeded the corresponding world average values. The elevated concentration of $^{40}$K is mainly due to the salinity intrusion, usage of fertilizers and agricultural runoff, and migration of waste effluents along the riverbanks. Being the pioneering comprehensive research on the Bangladesh side of the Sundarbans, this study forms a baseline radioactivity for the Sundarbans before the commissioning of the Rooppur Nuclear Power Plant in Bangladesh.

## 1. Introduction

Natural background radiation is ubiquitous in our dwelling environment. It arises mostly from the Naturally Occurring Radioactive Materials (NORMs) such as $^{238}$U, $^{232}$Th, and $^{40}$K and their decay products such as $^{226}$Ra, $^{222}$Rn, etc. NORMs are widespread on the Earth's surface and vary depending on the geological formations of soils, sediments, rocks, water, vegetation, etc. Increased use of ionising radiation or anthropogenic radioactivity may pose non-

**Funding:** The author(s) received no specific funding for this work.

**Competing interests:** The authors have declared that no competing interests exist.

negligible threats to living beings; therefore, radio-ecological assessment must be carried out to gather information on the radiological hazards to human and animal life in this present era of nuclear technology. The coastal environment and other aquatic ecosystems may provide notable information on radiological contamination because they facilitate the migration and accumulation of radioactive materials [1]. The present study site, Sundarbans, is a well-known coastal-aquatic ecosystem whose soils and sediments are also used to build homes and shelters for the nearby inhabitants.

The Sundarbans, located in the southwestern region of Bangladesh, is the largest block of tidal halophytic mangrove forest in the world, lying on the delta of Meghna, Brahmaputra, and Padma rivers on the Bay of Bengal. The forest comprises about 200 islands, divided by about 400 interconnected tidal rivers, creeks, and canals. In 1997, UNESCO declared 'The Sundarbans' a World Heritage Site owing to its species of biodiversity [2]. The Sundarbans is also the center of various economic activities such as timber extraction, honey collection, and fishing [3]. The Sundarbans provide a land-sea interface and seawater-freshwater interface that is home to a diverse ecosystem and provides a unique connection of different atmospheres, geologies, lithosphere, and hydrosphere. It has been reported that the Sundarbans is affected by highly diverse agricultural and aquaculture activities, which involve the usage of fertilizers and other chemicals, as well as soil erosion and wastewater runoff into the Sundarbans from these sites [4]. Since The Sundarbans is a vital site for agriculture and fishing, raised levels of radioactivity may cause by the influx of fertilizers and chemical pollution, which might affect the thousands of animal species living in the forest, and the bio-magnification of radioactivity may also affect the human population via the food chain. In the Indian region of the Sundarbans, the average activity of $^{40}$K (532–1043 Bqkg$^{-1}$) was reported to be more than twice as high as the global average of 420 Bqkg$^{-1}$ [5]. The authors hypothesised that the accumulation of upstream wastes and undesirable effluents along this coastal zone, together with rising salinity and the usage of fertilisers to boost crop productivity, may have collectively inficted K content. As the implications are comparable, it is necessary to do a similar radioactivity measurement in the Bangladesh part of the Sundarbans.

Moreover, Southern Bangladesh is experiencing major industrial developments, including the construction of 9 power plants. Of these, 6 projects are located in 3 locations surrounding the Sundarbans. Among these power plants, the Rampal Coal-based Thermal Power Plant poses the greatest concern, as it is located only 4km outside of the Ecologically Critical Area (ECA) on the east bank of the Pashur River. Furthermore, the Pashur River will be used for the transportation and trans-shipments of coal for the Rampal Power Plant, which poses the risk of coal spillage in the river. It has also been reported that 1–2 annual shipping accidents occur in the Sundarbans, which mainly involve coal and fertilizer shipping [6, 7]. In addition, The Rooppur Nuclear Power Plant is due to commissioning in 2023–24, which will use the Padma River water as its tertiary coolant. Since the Sundarbans lies at the delta of Padma River, the releases, if any, from the power plant might affect the overall radioactivity level of the waterbodies, sand, and sediments of the Sundarbans. Considering all of these foreseeable events, it is necessary to obtain the baseline data of NORMs in the Sundarbans areas, to monitor any radiological changes in the future due to natural or anthropogenic activities.

A few studies were reported on radiation levels of the mangrove forests in Pernambuco, Brazil, where maximum level of $^{40}$K was found to be 1338 Bqkg$^{-1}$, and the authors concluded that this was due the influence of sediments and presence of granites [8]. Radioactivity studies conducted in the petrified wood forests in Egypt found levels of $^{238}$U (65.26±12.99 Bqkg$^{-1}$) exceeding the world average values, along with the presence of the artificial radionuclide $^{137}$Cs arising from the nuclear accident in Chernobyl, and from the deposition from nuclear weapon tests in the neighbouring countries [9]. Very high concentrations of $^{232}$Th were found in three

Norwegian forests due to their proximity to an active volcano and the complex mixture of heavy mineral salts present there [10]. In the Indian Sundarbans mangrove forest, large amounts of 40K were discovered (maximum 866 Bqkg$^{-1}$), mostly due to the continuous deposition and erosion of silt, sediment, and other organic matter [1], also low levels of NORMs were discovered in the Krusadai Island Mangrove as a result of the region's low concentration of radiation-bearing minerals [11], and nitrogen fertilisers, a source of $^{40}$K in the environment, are widely used to improve soil nitrogen balance for agricultural development [12]. Moreover, it was found that the activity of $^{238}$U and $^{232}$Th had decreased from pre-tsunami data (25.9± 15.8 Bqkg$^{-1}$ and 65.1±34.5 Bqkg$^{-1}$) to post-tsunami data (12.2± 4.2 Bqkg$^{-1}$ and 11.7±5.0 Bqkg$^{-1}$) [13]. Although, the conditions are similar in the Sundarbans of Bangladesh only two studies have been found to date in the Sundarbans of Bangladesh site, one of which reported levels of NORMs were below the world average values in sediment samples [14] and the other study assessed the number of trace elements in sediments samples revealing, using various environmental contamination indices, that the sediments of the region have moderate to severely contaminated levels of Cd and moderate to low levels of As, Sb, Th, and U [15], but no systematic study has been carried out to date, to the best of our knowledge, to measure the radioactivity of soil in the Sundarbans. Therefore, this present study aims to measure the prevailing concentration of NORMs in the soils of Sundarbans, the first of its kind, to assess the impact of human activities (coal fired power plant, agriculture, uncontrolled fishing activities) on the Sundarbands ecosystem by investigating the radioactivity distribution. This study also aims to provide baseline data which is important due to the recent commissioning of a nuclear power plant and several thermal power plants surrounding the Sundarbans.

## 2. Methodology

### 2.1 Study area

The area is located in the southwestern part of the deltaic coastal plain region of Bangladesh (Fig 1A). Physiographically, the mapped area belongs to the physiographic units of the Ganges Tidal Floodplain and Sundarbans [16]. The topography is mainly flat with gentle relief (elevation ranges from 0.5 to 3.79 m above mean sea level), formed under fluvio-tidal conditions [15]. The sediments (Holocene terrain) mainly consist of an admixture of clay, silt, and fine sand. The covering morphological units are upper tidal and lower tidal plains (Fig 1B–1D). The largest mangrove forest (Sundarbans) in the world falls under the investigated area, specially Sarankhola Range and Chandpai Range. The region experiences a humid tropical monsoon climate. The average rainfall in the location from 2006–2011 was about 2024.8 mm/year. The area has four types of soils: Peat soils, Acid Sulphate Soils, Calcareous Brown Floodplain Soils and Calcareous Grey Floodplain Soils [16, 17].

The Landsat satellite image of 2014 was downloaded from the website http://glovis.usgs.gov and modified for use in this study. The layer stack of the image was performed by Erdas Imagine 2014 software. The visual image interpretation was carried out by ArcMap 10.2 to delineate the different geomorphic units of the area (Fig 1B–1D) as well as subsequent field checking according to our previous research [18–20].

### 2.2 Sampling and preparation procedure

A total of 30 topsoil samples were collected from the area around the Sundarbans (15 from Karamjal and 15 from Harbaria) in December 2021, and a global positioning system was used to record each location. The samples were collected following the systematic random sampling technique [21].

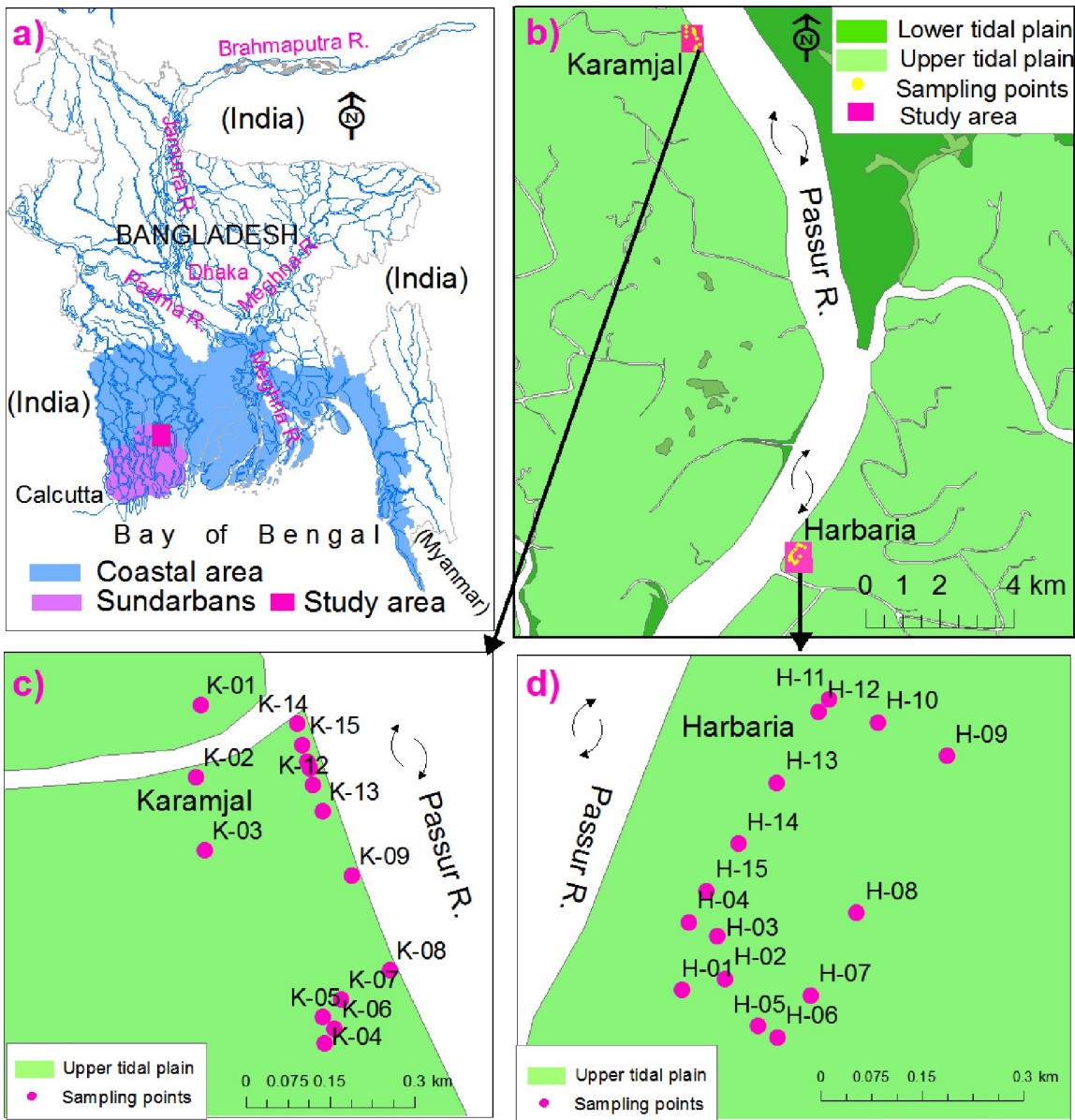

**Fig 1.** a) Map representing Bangladesh and its surrounding areas, coastal area of Bangladesh, Sundarbans Mangrove forest (modified after [18]) and locations of the study areas; b) study areas surrounding geomorphology; c) geomorphology of Karamjal and sampling points; d) geomorphology of Harbaria and sampling points.

After removing extraneous components like roots, pebbles, and plant matter, along with other impurities, the samples were homogenously mixed. Each sample, weighing between 0.5 and 1 kg, then immediately stored in airtight, clean zip-lock polyethylene bags, adequately labeled. The samples were transported to the sample preparation room of the Health Physics Division of Atomic Energy Centre Dhaka (AECD) for further processing. The samples were first dried under the sunlight; then the samples were dried carefully approximately at 105˚C-110˚C using an oven for four days, subsequently crushed with mortar and pestle, homogenized, and screened with a test sifter of opening 425 μm. Then all samples were then put into radon-impermeable, airtight Marinelli beakers (EG &G, Ortec). Then they were kept for 30

days to ensure that $^{226}$Ra and $^{232}$Th were in secular equilibrium with short-lived daughter products.

## 2.3 Measurement procedures and data analysis

Using a high-resolution coaxial HPGe gamma-ray spectrometer, the activity concentrations of radionuclides within the samples were determined. The detector was contained in a cylindrical lead shielding device with a sliding cover and a fixed bottom to reduce noise interference from the environment. A detector's ability to differentiate or discriminate the presence of two gamma rays that are closely spaced in energy is defined as the energy resolution of the detector. With a relative efficiency of 30%, it was found that the energy resolution of the 1.33 MeV energy peak for $^{60}$Co was 1.67 keV at full-width half-maximum (FWHM).

An empty sealed beaker was counted in the same way and with the same geometry as the samples before the sample measurement to figure out the background distribution in the area surrounding the detector. To reduce the degree of uncertainty in the net counts, an equal counting duration of 30000s for background and sample measurement was selected. Using the characteristic gamma lines of 241 keV, 295 keV, and 351 keV for $^{214}$Pb and 609 keV, 1120 keV, and 1764 keV for $^{214}$Bi, the activity concentration of $^{226}$Ra was estimated. On the other hand, the characteristic gamma lines 911 keV and 969 keV for $^{228}$Ac, were used to determine the $^{232}$Th activity concentration [22, 23]. Using 1460 keV and 661 keV gamma line, the radioactivity of $^{40}$K and $^{137}$Cs were estimated, respectively. The following equation [24] was used to determine each radionuclide's radioactivity concentration:

$$A_i = \frac{Z}{\varepsilon \times \rho_\gamma \times w} \tag{1}$$

Where $A_i$ is the specific activity in Bqkg$^{-1}$, Z is the net count rate per second (cps) = (samples cps- background cps), $\varepsilon$ is the HPGe detector's counting efficiency at the specific gamma-ray energy, $\rho_\gamma$ represents gamma-ray emission probability, and $w$ is the mass of the sample in kilograms (kg). The minimal detectable activity concentration (MDA) for the gamma-ray measurement system method was calculated using Eq (2) as stated in [25]:

$$MDA = \frac{K_\alpha \times \sqrt{B}}{\varepsilon \times \rho_\gamma \times T \times w} \tag{2}$$

where $K_\alpha$ is the statistical coverage factor, with a value of 1.64 (at the 95% confidence level), $B$ is the number of background counts for the relevant radionuclide, $T$ is the counting time, and $\rho_\gamma$ and $w$ (in kg) have the same usual meaning as in Eq (1). The MDAs for $^{226}$Ra, $^{232}$Th, and $^{40}$K were determined to be 0.35 Bq/kg, 0.64 Bq/kg, and 2.2 Bq/kg, respectively.

Using the uncertainty propagation law of the relevant quantities represented in Eq (2), the uncertainty of the measured radioactivity was determined. Eq (3) expressed the mathematical formulation for calculating the uncertainty of the determined radioactivity [26, 27]

Combined Standard Uncertainty

$$= A_i \times \sqrt{(\frac{u(N)}{N})^2 + (\frac{u(T)}{T})^2 + (\frac{u(\rho_\gamma)}{\rho_\gamma})^2 + (\frac{u(w)}{w})^2 + (\frac{u(\varepsilon)}{\varepsilon})^2} \tag{3}$$

The sample counts, counting time, gamma-ray emission probability, sample weight, and counting efficiency are represented by the letters N, T, $\rho_\gamma$, w, and $\varepsilon$, respectively. The calculated uncertainty of the relevant radionuclides varies from 10 to 15%.

## 2.4 Energy and efficiency calibration

The accuracy of the measured data largely depends on the energy and efficiency calibration of the detector, which must be carried out with extreme care. The detector's energy calibration was performed using common point sources like $^{22}$Na, $^{57}$Co, $^{60}$Co, $^{88}$Y $^{133}$Ba, $^{137}$Cs, $^{152}$Eu, etc. The percentage of radiation quanta (particles or photons) that a detector can detect out of all the radiation quanta that a source emits is known as the detector's efficiency. The IAEA standards (RGU-1, RGTh-1, and RGK-1) were used to determine the efficiency of the detector, which are Certified Reference Materials (CRM) and contain natural radionuclides from $^{238}$U-series, $^{232}$Th-series, and $^{40}$K, whose certified activity concentrations were 4940 ± 15 Bqkg$^{-1}$, 3250 ± 45 Bqkg$^{-1}$,14000 ± 200 Bqkg$^{-1}$ [28]. Besides, efficiency data was also checked by a standard source which was made by combining $^{152}$Eu of known activity (Liquid form, 900 Bq activity) with the $Al_2O_3$ matrix and manufactured in the same containers as the samples.

## 2.5 Radiological hazard parameters

**Radium equivalent activity.** The external and internal dose from $^{222}$Rn and its daughter are correlated by the radium equivalent activity, $Ra_{eq}$ (Bqkg$^{-1}$). The $Ra_{eq}$ was determined using Eq (4) to compare the combined radiological effect of $^{226}$Ra, $^{232}$Th, and $^{40}$K in the materials. For safe use, the maximum $Ra_{eq}$ value must be lower than 370 Bqkg$^{-1}$ [29].

$$Ra_{eq} = C_{Ra} + 1.43C_{Th} + 0.077C_K \tag{4}$$

$C_{Ra}$, $C_{Th}$, and $C_K$ represent the activity concentration of $^{226}$Ra, $^{232}$Th, and $^{40}$K in Bqkg$^{-1}$, respectively.

**The absorbed dose rate in air and annual effective dose evaluation.** The external absorbed dose rate, $D_{out}$, to the public's exposure due to the released gamma rays from the studied material at 1 m above the ground was calculated using the following Eq (5) [30]

$$D_{out} = 0.427C_{Ra} + 0.662C_{Th} + 0.0432C_K \tag{5}$$

$D_{out}$ represents the external absorbed dose rate in (nGy/h) due to gamma-ray exposure. Because human beings spend much more time indoors than outside, indoor exposure becomes more significant. Besides this, earth crust-derived products such as brick, sand, cement, paints, tiles, etc., are widely utilized in the construction of dwellings; therefore, assessing the indoor exposure is critical, and Eq (6) is used to calculate it [31]

$$D_{in} = 1.4D_{out} \tag{6}$$

The assessed indoor and outdoor exposures can be used to calculate the annual effective doses of $E_{in}$ and $E_{out}$. To accomplish this, the absorbed dose rate in the air was converted to the effective dose received by an adult using a conversion factor of 0.7 Sv/Gy [25]. Furthermore, because people spend roughly 80% of their time indoors and 20% outdoors, the values 0.8 and 0.2 for the indoor and outdoor occupancy factors are used to calculate the representing dosage. Thus, by using Eqs (7) and (8), the annual effective doses $E_{in}$ (mS/y) and $E_{out}$ (mSv/y) were calculated [30, 32].

$$E_{in}\left(^mSv/_y\right) = D_{in}\left(^nGy/_h\right) \times \left(8760^h/_y \times 0.7^Sv/_Gy \times 0.8\right) \times 10^{-6} \tag{7}$$

$$E_{out}\left(^mSv/_y\right) = D_{out}\left(^nGy/_h\right) \times \left(8760^h/_y \times 0.7^Sv/_Gy \times 0.2\right) \times 10^{-6} \tag{8}$$

### External hazard ($H_{ex}$) and internal hazard ($H_{in}$) indices evaluation

Using the external and internal hazard indices, the permissible equivalent dose should be lined up with a restricted value. Building materials should have a value of $H_{ex}$ that is less than or equal to unity to reduce the radiation dosage [25]. By using Eq (9) external hazard index ($H_{ex}$) can be calculated [30].

$$H_{ex} = \frac{C_{Ra}}{370} + \frac{C_{Th}}{259} + \frac{C_K}{4810} \tag{9}$$

Regarding the internal health risk, a quantitative index ($H_{in}$) known as the internal hazard index is provided by Eq (10) [33, 34].

$$H_{in} = \frac{C_{Ra}}{185} + \frac{C_{Th}}{259} + \frac{C_K}{4810} \tag{10}$$

### 2.6 Spatial distribution of different parameters

For interpolation of some derived data like radium equivalent activity, absorbed dose rate and annual effective dose in unsampling locations within the study area, interpolation was carried out using ArcGIS 10.2 software. The inverse distance weighting (IDW) technique was applied to interpolate the value of a variable at unmeasured sites from observations of its values at nearby locations according to our previous study [35, 36].

## 3. Results and discussion

Table 1 lists the activity concentrations in 30 soil samples taken from 15 distinct points in each Karamjal (K) and Harbaria (H) locations:

Almost all the values of $^{40}$K (except samples 5 and 6 in Karamjal), very few values of $^{232}$Th (Sample 3 in Karamjal and Sample 14 in Harbaria), and only one value of $^{226}$Ra (Sample 3 in Karamjal) show higher than the population-weighted world average values of 420, 45, 32 for $^{40}$K, $^{232}$Th and $^{226}$Ra respectively [32] in the studied soil samples. No artificial radionuclides were detected in the measured samples. The anthropogenic $^{137}$Cs were not found in the soil samples collected from the Sundarbans. The minimum detectable activity (MDA) of $^{137}$Cs is 1.54 Bqkg$^{-1}$. The highest activity of $^{226}$Ra, $^{232}$Th, and $^{40}$K obtained in the present study is 35, 50, and 660 Bqkg$^{-1}$, respectively, which exceeds the population-weighted world average values of 32, 45, 420 for $^{226}$Ra, $^{232}$Th, and $^{40}$K by 9%, 11%, and 57%. The order of natural radionuclide activity concentration was $^{40}$K>$^{232}$Th>$^{226}$Ra. In all terrestrial environments of the earth's crust, $^{40}$K is naturally abundant [37] and is a well-known primary weathering product [38]. Due to its high water solubility, $^{226}$Ra may experience surface runoff on muddy terrain, whereas $^{232}$Th sticks to the soil more due to its limited geochemical mobility [39]. Additionally, the lithosphere naturally contains three times as much thorium as uranium or radium [40]. Each location in the world has different geological and topographical conditions, which affect radioactivity in soils [41–43]. Variations in soil activity levels can be attributed to factors such as the soil-to-water ratio, the rate and amount of rainfall, soil drainage, site characteristics, and other environmental variables like meteorological conditions, soil use patterns, fertilizer use, evaporation, etc. [44]. Additionally, the radionuclides' chemical characteristics have a significant impact on how they migrate. The radioactivity levels of the examined soil are also influenced by the weathered components of the nearby deposited rocks [45]. The specific levels depend on the type of rock the soil is made of. Igneous rocks, like granite, have higher radiation levels, while sedimentary rocks have lower levels [46–51]. Low levels of $^{226}$Ra in the

**Table 1. Concentrations of $^{226}$Ra, $^{232}$Th, and $^{40}$K in soil samples collected from Karamjal and Harbaria.**

| Sample ID | Longitude (N) | Latitude (E) | $^{226}$Ra (Bqkg$^{-1}$) | $^{232}$Th (Bqkg$^{-1}$) | $^{40}$K (Bqkg$^{-1}$) |
|---|---|---|---|---|---|
| K-01 | 22.4286˚ | 89.59073˚ | 25±3 | 36±7 | 420±50 |
| K-02 | 22.42745˚ | 89.59065˚ | 26±3 | 35±7 | 500±60 |
| K-03 | 22.42628˚ | 89.5908˚ | 35±4 | 45±9 | 660±72 |
| K-04 | 22.42318˚ | 89.59288˚ | 29±3 | 36±7 | 490±58 |
| K-05 | 22.4236˚ | 89.59285˚ | 14±2 | 36±6 | 390±46 |
| K-06 | 22.42342˚ | 89.59305˚ | 23±2 | 31±6 | 370±44 |
| K-07 | 22.42388˚ | 89.59317˚ | 31±4 | 44±8 | 530±63 |
| K-08 | 22.42435˚ | 89.594˚ | 29±3 | 40±7 | 510±61 |
| K-09 | 22.42588˚ | 89.59335˚ | 34±4 | 39±7 | 630±69 |
| K-10 | 22.4276˚ | 89.59262˚ | 26±3 | 31±5 | 460±55 |
| K-11 | 22.4277˚ | 89.59257˚ | 22±3 | 30±5 | 500±60 |
| K-12 | 22.42732˚ | 89.59267˚ | 22±3 | 34±6 | 490±58 |
| K-13 | 22.4269˚ | 89.59283˚ | 26±3 | 30±5 | 470±56 |
| K-14 | 22.42832˚ | 89.5924˚ | 25±3 | 31±5 | 490±58 |
| K-15 | 22.42797˚ | 89.59248˚ | 27±3 | 34±6 | 500±60 |
| | | Average | 26±3 | 33±6 | 494±25 |
| | | Range | 14–35 | 30–45 | 370–660 |
| H-01 | 22.29905˚ | 89.61578˚ | 26±3 | 37±7 | 510±61 |
| H-02 | 22.2992˚ | 89.6164˚ | 29±3 | 44±8 | 560±67 |
| H-03 | 22.29977˚ | 89.61628˚ | 26±3 | 41±7 | 540±64 |
| H-04 | 22.29995˚ | 89.61587˚ | 22±3 | 38±5 | 540±64 |
| H-05 | 22.29857˚ | 89.61687˚ | 28±3 | 40±8 | 530±63 |
| H-06 | 22.29842˚ | 89.61715˚ | 30±3 | 43±8 | 540±64 |
| H-07 | 22.29898˚ | 89.61762˚ | 28±3 | 36±5 | 580±69 |
| H-08 | 22.30008˚ | 89.61828˚ | 20±2 | 35±5 | 520±62 |
| H-09 | 22.30218˚ | 89.61957˚ | 24±3 | 34±5 | 540±64 |
| H-10 | 22.30263˚ | 89.61858˚ | 22±3 | 35±5 | 570±68 |
| H-11 | 22.30293˚ | 89.61788˚ | 29±3 | 41±7 | 530±63 |
| H-12 | 22.30277˚ | 89.61773˚ | 30±3 | 38±6 | 540±64 |
| H-13 | 22.30182˚ | 89.61713˚ | 26±3 | 33±5 | 550±66 |
| H-14 | 22.301˚ | 89.61658˚ | 31±4 | 50±9 | 520±62 |
| H-15 | 22.30037˚ | 89.61613˚ | 25±3 | 38±7 | 530±63 |
| | | Average | 27±3 | 39±5 | 540±64 |
| | | Range | 20–31 | 34–50 | 510–580 |

collected soil are due to the absence of interaction with igneous rocks and uranium-rich minerals such as apatite, zircon, etc. [40]. In soil that contains much monazite, the concentration of $^{232}$Th is higher [52–54]. Comparatively low values of $^{232}$Th in the current study indicate the absence of monazite-bearing minerals.

Large portions of the Sundarbans have been recovered during the past few decades for use in agriculture and settlement [5, 12]. In some circumstances, fertilizers are frequently employed to enhance the nitrogen balance of soil for agronomic growth, which is a source of $^{40}$K in the environment [5, 12, 55]. The majority of the soils of the Sundarbans are heavy-textured, brownish/greyish black silty or clayey in composition, with the availability of swampy soils near the sea [21]. The $^{40}$K concentration of soils and sediments is typically high since it is indigenous to this coastal region without any such influence from fertilizers [56, 57]. Several cargo catastrophes involving coal fly ash, fertilizer, and oil have recently happened along the

Sela and Poshur rivers in Bangladesh's Sundarbans [15]. There are worries about the ecological catastrophe that occurred in the Sundarbans due to the Sela and Poshur rivers' connections to the tiny creeks there. The mangrove belt in the Sundarbans estuary is very dynamic, with consistent deposition and erosion of silt and sediment from various interconnected rivers, creeks, channels, etc. [1]. The Mongla port city, which is close to the Sundarbans, is home to a number of oil, petroleum, and cement factories. Therefore, throughout the Sundarbans estuary, there is a substantial risk of buildup from home and agricultural effluents, industrial trash, etc. Overall, it can be concluded that the Sundarbans have a high native $^{40}K$ content, and factors such as rising salinity, the use of fertilizers to boost crop yields, and the buildup of upstream waste and undesirable effluents along this coastal zone may have contributed to high $^{40}K$ content.

Table 2 provides a comparative analysis of the mean activity concentrations of $^{238}U$, $^{232}Th$, and $^{40}K$ in the present study with those of other studies analyzing soil samples collected from mangrove forests across the world. The $^{40}K$ levels were found high in some mangrove forests, such as in the Rio Formoso mangrove (RFM) [8] located in Pernambuco, Brazil (Maximum

**Table 2. Radiological data for $^{238}U$, $^{232}Th$, and $^{40}K$ associated with soil samples collected from mangrove forests from different countries.**

| Sl No | Region of Study | | Mean (Bqkg$^{-1}$) | | | Methodology | Reference |
|---|---|---|---|---|---|---|---|
| | | | $^{226}Ra$ ($^{238}U$) | $^{232}Th$ | $^{40}K$ | | |
| 1 | Pernambuco, Brazil | The Rio Formoso mangrove | 21 | 43($^{228}Ra$) | 869 | HPGe detector (40% *RE) | [8] |
| | | The Chico science mangrove | 24 | 41 ($^{228}Ra$) | 414 | | |
| 2 | Sundarbans Mangrove, India | | 48.7 | 58.5 | 866.2 | HPGe detector (50% RE) | [1] |
| 3 | Krusadai Island Mangrove, Gulf of Mannar, India | | BDL | BDL to 27.81 ± 8.9 | BDL to 413.13 ± 49.6 | NaI (Tl) detector | [11] |
| 4 | Pichavaram Mangroves, South East Coast of India (Sediment) | (Post-Tsunami) | 12.2±4.2 | 11.7±5.0 | 265±133.8 | HPGe detector | [13] |
| | | (Pre-Tsunami) | 25.9±15.8 | 65.1±34.5 | 190.7±32.5 | HPGe detector | |
| 5 | Petrified wood forest, El-Qattamia, Cairo, Egypt | | 65.26±12.99 | 23.66±0.95 | 146.33±1.5 | HPGe detector (50% RE) | [9] |
| 6 | Norway | Fen | 126 | 2737 | - | HPGe detector | [10] |
| | | Bolladalen | 165 | 6835 | - | | |
| | | Torsnes | 11 | 664 | - | | |
| 7 | Bhawalgahr forest, Gazipur district, Bangladesh | (0-5cm depth) | 63.5 ± 6.8 | 104.5 ± 8.9 | 433.9 ± 96.4 | HPGe detector (20% RE) | [58] |
| | | (5–15 cm depth) | 58.5 ± 5.6 | 98.2 ± 8.9 | 491.7 ± 101.3 | | |
| | | (15–30) cm depth | 60.8 ± 5.6 | 97.8 ± 8.6 | 508.5 ± 100.9 | | |
| 8 | Shamnagarupazila, Satkhira, south-west of Sundarbans, Bangladesh | Surface soil (5–20 cm depth) | 35.59±3.94 | 37.69±3.92 | 398.73±24.19 | HPGe detector | [59] |
| | | Deep soil (130–150 cm depth) | 40.65±3.92 | 50.61±2.78 | 476.62±24.51 | | |
| 9 | Sundarbans Mangrove, Bangladesh | | 23±2 to 50±10 (Range) | 34±5 to 79±6 (Range) | 248±46 to 561±58 (Range) | HPGe detector | [14] |
| 10 | Sundarbans Mangrove, Bangladesh | Karamjal | 26±3 | 33±6 | 494±25 | HPGe detector (30% RE) | Current study |
| | | Harbaria | 27±3 | 39±5 | 540±64 | | |

*RE = Relative Efficiency

$^{40}$K level 1338 Bqkg$^{-1}$), in Sundarbans Mangrove, India [1] (Maximum $^{40}$K level 1043 Bq/kg). This behavior of high $^{40}$K in RFM mangroves was linked to the strong marine influence bringing sediments enriched with $^{40}$K on the mangroves, as well as the fact that the RFM location is surrounded by granites holding around 7% K$_2$O [8]. Nabanita Naskar et al. [1] explained that the possible reason for the high level of $^{40}$K in the Indian Sundarbans is that the tidal mangrove belt in the Indian Sundarbans is highly dynamic due to the constant deposition and erosion of silt, sediment from the many rivers, creeks, channels, etc. that flow into it.

In a fascinating comparison of pre-and post-tsunami radioactivity in sediment samples from the Pichavaram mangroves, Satheeshkumar et al. [13] found that the activity of $^{238}$U and $^{232}$Th had decreased from pre-tsunami data (25.9± 15.8 Bqkg$^{-1}$ and 65.1±34.5 Bqkg$^{-1}$) to post-tsunami data (12.2± 4.2 Bqkg$^{-1}$ and 11.7±5.0 Bqkg$^{-1}$). The hydrodynamics of the tsunami waves, which removed vast quantities of decades-old beach sediments, is a crucial factor in the low level of radioactivity in Pichavaram. In addition, it spread a substantial layer of black sand. As a result, background radiation levels in the environment are reduced [13]. In the Krusadai Island Mangrove, Gulf of Mannar, India, low concentrations of $^{238}$U, $^{232}$Th, and $^{40}$K were found due to the low concentration of radiation-bearing minerals in the study area, as well as the coastal geography and the direction of the currents, the concentration of uranium- and thorium-bearing minerals is reduced [11]. The Norwegian sites of Fen and Bolladalen had extremely high $^{232}$Th concentrations [48] because the whole Fen Complex was an active volcano suggests that different ores were present at varying depths and that there were heterogeneous rocks present that included mixes of several minerals and different elements. On the other hand, Bolladalens was seen as a representation of legacy NORM and unaltered $^{232}$Th-rich sites, respectively, due to its complex mixture of rodbergite and rauhaugite.

The soil samples collected from three different depths (0–55, 5–15, and 15–30 cm) were analyzed to observe the horizontal as well as vertical distribution of the radioactivity concentration in the Bhawalgahr forest, Gazipur district, Bangladesh [58]. The average activity concentrations of $^{226}$Ra, $^{232}$Th, and $^{40}$K for all the samples were higher than those of the worldwide average values, which may be attributed to the area's geological characteristics. The vertical distribution of $^{232}$Th and $^{40}$K showed a decreasing tendency of activity concentration with depth, whereas no particular trend with depth was observed for $^{226}$Ra in that study. The results of the current study are in close agreement with earlier studies conducted globally due to the close geological coherence, such as in Bhawalgahr forest, Gazipur district, Bangladesh [49]; Shamnagarupazila, Satkhira, south-west of Sundarbans, Bangladesh [50]; Sundarbans mangrove, Bangladesh [14]; and Sundarbans mangrove; India [1].

The radium equivalent activity, absorbed dose rate (outdoor and indoor), external and internal hazard index, and external and internal annual effective dose values are reported in Table 3, and the spatial distribution of different parameters in Harbaria and Karamjal are represented in Figs 2 and 3.

All values of radium equivalent activity are far below the recommended limit of 370 Bqkg$^{-1}$ [60]. Some of the values of the outdoor absorbed dose rate in Karamjal (Samples 3,7–9) and most of the values of the outdoor absorbed dose rate in Harbaria (Samples 2,3,5–7,11–12,14) exceeded the population-weighted average outdoor absorbed dose rate in air in the world which is 59 nGyh$^{-1}$ [32]. In the case of indoor absorbed dose rate, almost all the values are within the population-weighted average indoor absorbed dose rate in air in the world, which is 84 nGyh$^{-1}$ [32]. The values of external and internal hazard index are less than unity in all the samples that indicate the non-hazardous for human beings [32]. Some of the values (3–4,7–9) of the outdoor annual effective dose in Karamjal and almost all (except Samples 8,9) of the outdoor annual effective dose in Harbaria exceed the worldwide average outdoor annual effective dose due to terrestrial radiation, which is 0.07 mSvy$^{-1}$ [32]. Almost all the values (except Sample 3 in

**Table 3. Data for different radiological parameters in soil samples of the current study.**

| Sample ID | Radium Equivalent Activity (Ra$_{eq}$) Bqkg$^{-1}$ | Outdoor Absorbed Dose rate (D$_{out}$) nGyhr$^{-1}$ | Indoor Absorbed Dose rate (D$_{in}$) nGyhr$^{-1}$ | External Hazard Index (H$_{ex}$) | Internal Hazard Index (H$_{in}$) | Outdoor Annual Effective Dose (E$_{out}$) mSvy$^{-1}$ | Indoor Annual Effective Dose (E$_{in}$) mSv$^{-1}$ | Total Annual Effective Dose (E) mSv$^{-1}$ |
|---|---|---|---|---|---|---|---|---|
| K1 | 108.82 | 52.65 | 63.18 | 0.29 | 0.36 | 0.065 | 0.310 | 0.375 |
| K2 | 114.55 | 55.87 | 67.05 | 0.31 | 0.38 | 0.069 | 0.329 | 0.397 |
| K3 | 150.17 | 73.25 | 87.90 | 0.41 | 0.50 | 0.090 | 0.431 | 0.521 |
| K4 | 118.21 | 57.38 | 68.86 | 0.32 | 0.40 | 0.070 | 0.338 | 0.408 |
| K5 | 95.51 | 46.66 | 55.99 | 0.26 | 0.30 | 0.057 | 0.275 | 0.332 |
| K6 | 95.82 | 46.33 | 55.59 | 0.26 | 0.32 | 0.057 | 0.273 | 0.330 |
| K7 | 134.73 | 65.26 | 78.31 | 0.36 | 0.45 | 0.080 | 0.384 | 0.464 |
| K8 | 125.47 | 60.90 | 73.07 | 0.34 | 0.42 | 0.075 | 0.358 | 0.433 |
| K9 | 138.28 | 67.55 | 81.06 | 0.37 | 0.47 | 0.083 | 0.398 | 0.481 |
| K10 | 105.75 | 51.50 | 61.80 | 0.29 | 0.36 | 0.063 | 0.303 | 0.366 |
| K11 | 103.40 | 50.85 | 61.02 | 0.28 | 0.34 | 0.062 | 0.299 | 0.362 |
| K12 | 111.21 | 54.39 | 65.27 | 0.30 | 0.36 | 0.067 | 0.320 | 0.387 |
| K13 | 105.09 | 51.27 | 61.52 | 0.28 | 0.35 | 0.063 | 0.302 | 0.365 |
| K14 | 107.06 | 52.37 | 62.84 | 0.29 | 0.36 | 0.064 | 0.308 | 0.372 |
| K15 | 114.12 | 55.64 | 66.76 | 0.31 | 0.38 | 0.068 | 0.328 | 0.396 |
| Average | 115.21 | 56.12 | 67.35 | 0.31 | 0.38 | 0.069 | 0.330 | 0.399 |
| H1 | 118.18 | 57.63 | 69.15 | 0.32 | 0.39 | 0.071 | 0.339 | 0.410 |
| H2 | 135.04 | 65.70 | 78.84 | 0.36 | 0.44 | 0.081 | 0.387 | 0.467 |
| H3 | 126.21 | 61.57 | 73.89 | 0.34 | 0.41 | 0.076 | 0.362 | 0.438 |
| H4 | 117.92 | 57.88 | 69.45 | 0.32 | 0.38 | 0.071 | 0.341 | 0.412 |
| H5 | 126.01 | 61.33 | 73.60 | 0.34 | 0.42 | 0.075 | 0.361 | 0.436 |
| H6 | 133.07 | 64.60 | 77.52 | 0.36 | 0.44 | 0.079 | 0.380 | 0.460 |
| H7 | 124.14 | 60.84 | 73.01 | 0.34 | 0.41 | 0.075 | 0.358 | 0.433 |
| H8 | 111.52 | 54.84 | 65.80 | 0.30 | 0.36 | 0.067 | 0.323 | 0.390 |
| H9 | 114.20 | 56.08 | 67.30 | 0.31 | 0.37 | 0.069 | 0.330 | 0.399 |
| H10 | 115.94 | 57.19 | 68.63 | 0.31 | 0.37 | 0.070 | 0.337 | 0.407 |
| H11 | 128.44 | 62.42 | 74.91 | 0.35 | 0.43 | 0.077 | 0.367 | 0.444 |
| H12 | 125.92 | 61.29 | 73.55 | 0.34 | 0.42 | 0.075 | 0.361 | 0.436 |
| H13 | 115.54 | 56.71 | 68.05 | 0.31 | 0.38 | 0.070 | 0.334 | 0.403 |
| H14 | 142.54 | 68.80 | 82.56 | 0.38 | 0.47 | 0.084 | 0.405 | 0.489 |
| H15 | 120.15 | 58.73 | 70.47 | 0.32 | 0.39 | 0.072 | 0.346 | 0.418 |
| Average | 123.65 | 60.37 | 72.45 | 0.33 | 0.41 | 0.074 | 0.355 | 0.429 |

Karamjal) of the indoor annual effective dose are within the worldwide average indoor annual effective dose, which is 0.41mSvy$^{-1}$ [32]. As almost all the values of $^{40}$K and most of the outdoor absorbed dose rate in air and outdoor annual effective dose exceeds the corresponding population-weighted world average values, we can conclude that the study area is safe for short-term stay by the visitors but not radiologically safe (specially Harbaria) for long-term residence because the presence of such radionuclides at high concentrations can potentially result not only in harmful health effects, such as chronic lung diseases, anemia, and different cancers but also in genetic mutations that affect not only humans but also the entire biota [43, 61].

## 4. Conclusions

This study was carried out to evaluate the distribution of naturally occurring radioactive materials ($^{226}$Ra, $^{232}$Th, and $^{40}$K), as well as artificial radionuclides, and to calculate the associated

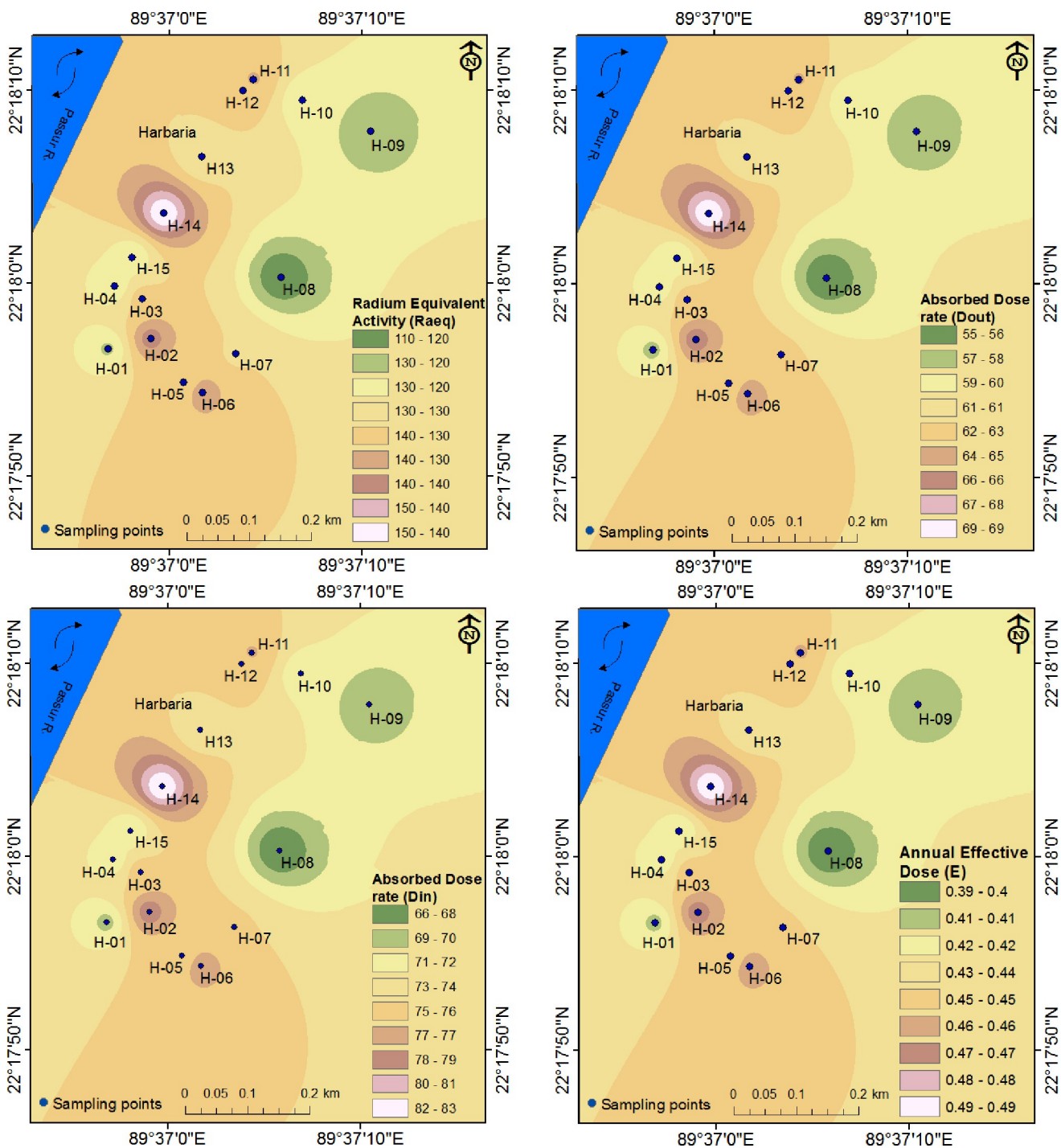

**Fig 2. Distribution maps of different parameters in the Harbaria area.**

radiological hazard parameters at Karamjal and Harbaria, the two primary tourist spots in the world's largest mangrove forest, Sundarbans.

The range of activity concentration of $^{226}$Ra, $^{232}$Th, and $^{40}$K in Karamjal was 14–35 Bqkg$^{-1}$, 30–45 Bqkg$^{-1}$, and 370–660 Bqkg$^{-1}$, respectively. In the Harbaria region, the values were 20–31

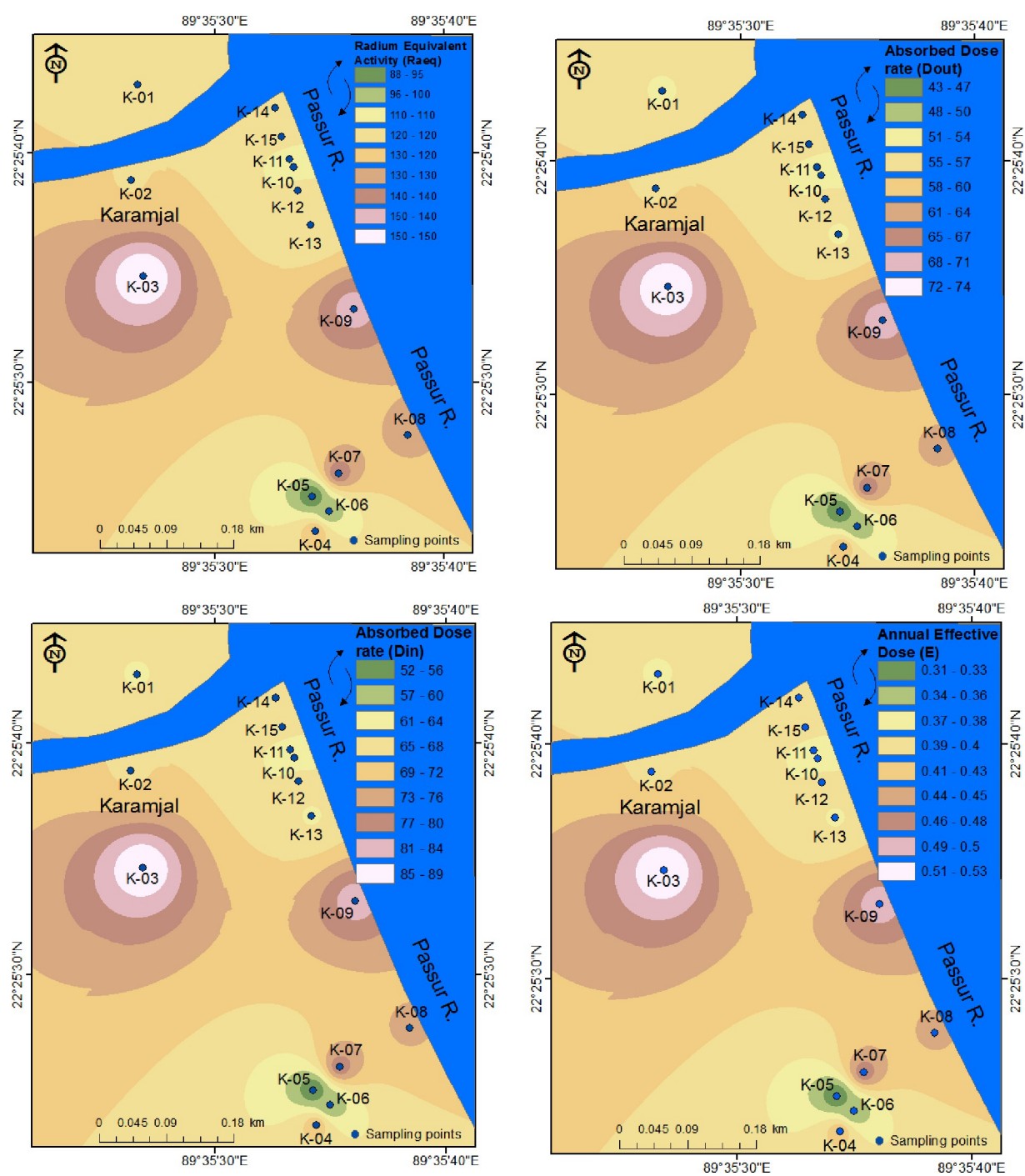

**Fig 3. Distribution maps of different parameters in the Karamjal area.**

Bqkg$^{-1}$, 34–50 Bqkg$^{-1}$, and 510–580 Bqkg$^{-1}$ for $^{226}$Ra, $^{232}$Th, and $^{40}$K, respectively. Most samples contain a higher $^{40}$K concentration than the population-weighted world average values, which is attributed to the natural abundance of potassium-bearing minerals, weathering process, fertilizer use, etc. In regard to anthropogenic activity, no radioactivity of $^{137}$Cs was detected in the collected soil samples.

Most of the hazard parameters are within the recommended safety limits, so the research areas are not posing a considerable hazard for short-term stay by visitors.

A few recommendations are proposed by this current research:

- A detailed survey is necessary for mapping the distribution of NORMs in the world's largest mangrove forest, Sundarbans, which will be essential reference data.

- Frequent monitoring is needed to evaluate the effect of the soon-to-be commissioned Rooppur Nuclear Power Plant in Bangladesh.

- The greater activity concentration of $^{40}$K indicates the existence of potassium-bearing mineral resources in the nearby areas.

- The outdoor effective dose in some regions is higher than the population-weighted average value, a significant concern for the local inhabitants; appropriate knowledge and mitigation strategies may need to be introduced.

## Author Contributions

**Conceptualization:** M. M. Mahfuz Siraz, Jubair A. M., M. S. Alam, Md. Bazlar Rashid, Mayeen Uddin Khandaker.

**Data curation:** M. M. Mahfuz Siraz, Z. Hossain.

**Formal analysis:** M. M. Mahfuz Siraz, Jubair A. M., M. S. Alam, Mayeen Uddin Khandaker, D. A. Bradley.

**Supervision:** Mayeen Uddin Khandaker, S. Yeasmin.

**Validation:** M. M. Mahfuz Siraz, Mayeen Uddin Khandaker.

**Visualization:** M. M. Mahfuz Siraz, Mayeen Uddin Khandaker.

**Writing – original draft:** M. M. Mahfuz Siraz, Jubair A. M., M. S. Alam, Md. Bazlar Rashid.

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
