## [Decision Letter · Decision Letter 0]

17 Nov 2022

PONE-D-22-26012

Elevated Levels of 40K in the tourist-captivating Karamjal and Harbaria sites of the World’s largest mangrove forest Sundarbans

PLOS ONE

Dear Dr. Siraz,

Thank you for submitting your manuscript to PLOS ONE. After careful consideration, we feel that it has merit but does not fully meet PLOS ONE’s publication criteria as it currently stands. Therefore, we invite you to submit a revised version of the manuscript that addresses the points raised during the review process.

We look forward to receiving your revised manuscript.

Kind regards,

Mohamad Syazwan Mohd Sanusi

Academic Editor

PLOS ONE

Journal Requirements:

5. Please ensure that you refer to Figure 2 in your text as, if accepted, production will need this reference to link the reader to the figure.

6. We note that Figure 1 in your submission contain [map/satellite] images which may be copyrighted. All PLOS content is published under the Creative Commons Attribution License (CC BY 4.0), which means that the manuscript, images, and Supporting Information files will be freely available online, and any third party is permitted to access, download, copy, distribute, and use these materials in any way, even commercially, with proper attribution. For these reasons, we cannot publish previously copyrighted maps or satellite images created using proprietary data, such as Google software (Google Maps, Street View, and Earth). For more information, see our copyright guidelines: http://journals.plos.org/plosone/s/licenses-and-copyright.

7. We note that Figure 2 includes an image of a participant in the study. 

Additional Editor Comments:

Dear authors, after careful evaluations, we would like to invite you to revise the submitted manuscript based on the attached comments

Reviewers' comments:

Reviewer's Responses to Questions

**Comments to the Author**

1. Is the manuscript technically sound, and do the data support the conclusions?

Reviewer #1: Partly

Reviewer #2: Partly

2. Has the statistical analysis been performed appropriately and rigorously? 

Reviewer #1: N/A

Reviewer #2: Yes

3. Have the authors made all data underlying the findings in their manuscript fully available?

Reviewer #1: Yes

Reviewer #2: Yes

4. Is the manuscript presented in an intelligible fashion and written in standard English?

Reviewer #1: Yes

Reviewer #2: Yes

5. Review Comments to the Author

Reviewer #1: Dear Authors,

The manuscript entitled Elevated Levels of 40K in the tourist-captivating Karamjal and Harbaria sites of the World’s largest mangrove forest Sundarbans is an interesting and well descried work. The aim is properly presented and motivated. Your results are important and potentially interesting for the international scientific community, especially as the described issue has an interdisciplinary manner.

The abstract suits well the manuscript content. The tables and figures are informative and well readable. Results dissuasion is properly described and justified with adequate literature references.

Nevertheless, there are some serious problems that need to be issued.

(order according to occurrence in the text)

1. Your work deal mostly with radiological issues (radiological hazard, dosimetry), not geology. That is why there is no need to give so much geological information in the methodology section, like detailed soils description. Especially you have not used these data in the results discussion and interpretation.

2. What about gamma radiation background influence? You have not mentioned anything that you extract background from the cps or N of your samples. It is crucial for the proper calculation of the results. Especially for 40K, it is always present in the background to the great extent. Did you do the background extraction? If yes, it definitely needs to be described in the text. Give all details like: way of measurements (empty detector, empty container), time of a single measurement, if time of background measurement was comparable with sample measurement time (btw. which you did not provide in the text), and level of the background. If background extraction has not been done it is necessary to do it! Otherwise, such results cannot be accepted for publication.

3. You have mentioned that there is no 137Cs in your samples. It means that you did a research on 137Cs. Meanwhile there is no description how it was done. No energy line or detection limit of 137Cs is provided.

4. Description of the detector efficiency determination is given but no information on the results. At least relative efficiency of the detector should be presented.

5. The number of decimal places in the results must be ordered. It is inconsistent and partially incorrect in the results. The result and the uncertainty of the result should have the same number of digits. If the result is in the hundreds, there is no need to give two decimal places; they are less than 1% of the result. General rule is that: two or three significant digits are enough.

6. In the discussion of the results, it is stated many times that the obtained results exceed the limits. But the degree of this excess is not specified. The reader must analyze the whole tables with results and look for reference values in the text. You can specify the percentage range by which the results exceed the accepted limits. As it currently stands, it is not clear whether these are serious surpluses or just a few percent. This can only be guessed on the basis of the recommendations given by the authors at the end of the text.

My other minor remarks are given as the comments in the text in attached pdf file.

Reviewer #2: The papers presents results of measurements of natural radioactivity (by means of HPGe gamma spectrometry) in some mangrove forest soil in Bangladesh. The studied area is not subject of many papers therefore I suggest to publish the results. However I have some objection regarding the methodology, the possible lack of secular equilibrium in Th series is not enough addressed. Also the calibration done using Eu-152 is not the state of the art method. I would not call any of result obtained in paper as “the elevated levels”. Typical levels for K-40 are in range of 140-800 Bq/kg, so all presented here results are in this range. This implies the reconsideration of discussion of results. Therefore my general recommendation is “major revision”.

My particular comments:

Row 19 and rows 171-171. One cannot measure directly Th-232 via gamma radiation. What is measured are Th series progenies (Ac-228 and Tl-208, depending which lines you took under consideration). Due to high risk of lack of equilibrium in open environment they might reflect the activities of Ra-228 (case of Ac-228 lines) or Th-228 (case of Tl-206 lines). The Th-228 and Ra-228 in fresh sediment samples could not be in equilibrium, so one should present the results from Ac-228 (which could be named “Ra-228”) and Tl-208 (which could be named “Th-228”) separately.

Row 28. Space between “etc.” and “Elevated”. The lack of space after dot or between the number and unit (for instance row. 66) repeats in many places in text, please check and correct all. BTW – there is no really elevated levels among the results.

Row. 153. Ra-226 and Ra-224 are parents, not daughters of mentioned Rn-222 and Rn-220, respectively. You cannot call them also “elements”, since the mass number is specified (nuclide, radionuclide, radioisotope or isotope are far more proper) .

Row. 154. The way how the equation 1 is written could be misleading. When I have looked on denominator I’ve seen “exp”, what is nonsense. With more careful look I realized, that “x” here is multiplication symbol. The “cps” in numerator is also not sure, if it is a product of three values or three character symbol. I suggest to re-write this formula avoiding many character symbols (instead pleas use the single character symbols with indexes) and please replace the “x” by dots. Please use the same style (italic) for symbols in all equations and in all description of them (incl. eq. 4 and 5). Please also note, that “mass” and “weight” are not the same things in physics, the proper term here is “mass”. Please correct this way all equations. The Eq. 3 is correct is you do not take into consideration any background, otherwise it is more complex and in case of K-40 you have to take into account the correction for spectrometer background.

Row 201. The usage of Eu-152 for efficiency calibration is not proper due to cascade transitions in decay of this isotope. One can do corrections for it using one of several methods. It looks that you have not done that correction. Without a cascade correction some of your results could by biased for even 10%.

Table 1. – the presentation of results is not proper. The data for Th series should be presented separately for Ac-228 and Tl-208 lines. In general the results should be presented in way: xab.c±y.z or xab±vw but never xab±fg.h. The rule is: present the uncertainty only in two meaning digits. The way of presentation of main part of result is driven by the precision of uncertainty. In particular 500±60 is OK, but 490±58.8 is not OK. The 33±3.63 is not OK as well. But 40±6 can be accepted.

6. PLOS authors have the option to publish the peer review history of their article (what does this mean?). If published, this will include your full peer review and any attached files.

Reviewer #1: No

Reviewer #2: No

---

## [Author Response · Author response to Decision Letter 0]

22 Dec 2022

Response to academic editor and reviewer(s) 

Respected Academic Editor and Reviewers,

Thank you so much for your comments. We are very grateful to you for the time and intelligence that you have shared with us. It is a great learning opportunity for us through these reviews. Our responses to the comments are highlighted in red colour in the revised manuscript.

Comments and Suggestions for Authors:

Academic Editor

Academic editor comment 1: Please ensure that your manuscript meets PLOS ONE's style requirements, including those for file naming. The PLOS ONE style templates can be found at 

Response: Thank you so much for your kind comments and appreciation of our study. All of your comments are carefully evaluated and revised in our revision accordingly. The revised manuscript has been prepared according to Plos One style.

Academic editor comment 2: In your Methods section, please provide additional information regarding the permits you obtained for the work. Please ensure you have included the full name of the authority that approved the field site access and, if no permits were required, a brief statement explaining why.

Response: We thank the editor for this comment. The authors are from Bangladesh Atomic Energy Commission, a government organization in Bangladesh. As the main and relevant government institution, we have the responsibility to obtain the baseline radioactivity prior to the operation of the country’s first nuclear power plant. On the other hand, no written permission is needed from any other authority since the studied soil samples were collected from a large area which is not a restricted area.

Academic editor comment 3: In your Data Availability statement, you have not specified where the minimal data set underlying the results described in your manuscript can be found…

Response: We appreciate the reviewer's feedback. All relevant data are within the manuscript.

Academic editor comment 4: We note that you have stated that you will provide repository information for your data at acceptance. Should your manuscript be accepted for publication, we will hold it until you provide the relevant accession numbers or DOIs necessary to access your data. If you wish to make changes to your Data Availability statement, please describe these changes in your cover letter, and we will update your Data Availability statement to reflect the information you provide.

Response: We thank the reviewer for this opportunity to clarify this issue. All relevant data are within the manuscript. We will write a cover letter regarding this issue. 

Academic editor comment 5: Please ensure that you refer to Figure 2 in your text as, if accepted, production will need this reference to link the reader to the figure.

Response: Figure 2 has been deleted from the revised manuscript.

Academic editor comment 6: We note that Figure 1 in your submission contain [map/satellite] images which may be copyrighted. All PLOS content is published under the Creative Commons Attribution License (CC BY 4.0), which means that the manuscript, images, and Supporting Information files will be freely available online, and any third party is permitted to access, download, copy, distribute, and use these materials in any way, even commercially, with proper attribution. For these reasons, we cannot publish previously copyrighted maps or satellite images created using proprietary data, such as Google software (Google Maps, Street View, and Earth). For more information, see our copyright guidelines: http://journals.plos.org/plosone/s/licenses-and-copyright.

Response: We appreciate the reviewer giving us a chance to explain this matter.

Reference has been provided in the Figure caption like this, "Figure 1: a) Map representing Bangladesh and its surrounding areas, coastal area of Bangladesh, Sundarbans Mangrove forest (modified after (19))". 

In the text, a separate section has been provided with appropriate reference (Page No. 5, Line No. 116), "2.2 Geomorphological map preparation: Landsat satellite image of 2014 was downloaded from the website http://glovis.usgs.gov and modified for use in this study. The layer stack of the image was performed by Erdas Imagine 2014 software. The visual image interpretation was carried out by ArcMap 10.2 to delineate the different geomorphic units of the area (Figure 1 b,c,d) as well as subsequent field checking according to our previous research [18]–[20].

For Figures 2 and 3, a separate section has been added in the revised manuscript(Page No. 9, Line No. 229), "2.7 Spatial distribution of different parameters: To examine the spatial distribution of different parameters, GIS (Geographic Information System) mapping and interpolation were carried out using ArcGIS 10.2 software and modified for use in this study. The inverse distance weighting (IDW) technique was applied to interpolate the value of a variable at unmeasured sites from observations of its values at nearby locations according to our previous study [37], [38]." 

Academic editor comment 7: We note that Figure 2 includes an image of a participant in the study. …..

Response: Figure 2 has been deleted from the revised manuscript.

Reviewer #1 comment: 

Reviewer 1, comment 1: Your work deal mostly with radiological issues (radiological hazard, dosimetry), not geology. That is why there is no need to give so much geological information in the methodology section, like detailed soils description. Especially you have not used these data in the results discussion and interpretation.

Response: We appreciate the opportunity to explain this matter to the reviewer. We fully agree with you. We have shortened 2.1: Study Area (Page No.4, Line No. 99) and the detailed soil description has been deleted from the revised manuscript.

Reviewer 1, comment 2: What about gamma radiation background influence? You have not mentioned anything that you extract background from the cps or N of your samples. It is crucial for the proper calculation of the results. Especially for 40K, it is always present in the background to the great extent. Did you do the background extraction? If yes, it definitely needs to be described in the text. Give all details like: way of measurements (empty detector, empty container), time of a single measurement, if time of background measurement was comparable with sample measurement time (btw. which you did not provide in the text), and level of the background. If background extraction has not been done it is necessary to do it! Otherwise, such results cannot be accepted for publication. 

Response: We thank the reviewer for this opportunity to clarify this issue. Yes, the background was measured during the experiment, and a detailed description of the background has been added in the revised manuscript (Page No.6, Line No. 147) like this, “An empty sealed beaker was counted in the same way and with the same geometry as the samples before the sample measurement to figure out the background distribution in the area surrounding the detector. To reduce the degree of uncertainty in the net counts, an equal counting duration of 30000s for background and sample measurement was selected. "

Reviewer1, comment 3: You have mentioned that there is no 137Cs in your samples. It means that you did a research on 137Cs. Meanwhile there is no description how it was done. No energy line or detection limit of 137Cs is provided. 

Response: We are grateful that the reviewer gave us the opportunity to address this. The energy line of Cs-137 has been added in the revised manuscript like this (Page No.6, Line No. 154), “Using the unique 1460 keV and 661 keV gamma line, the radioactivity of 40K and 137Cs was estimated, respectively.”

Reviewer1, comment 4: Description of the detector efficiency determination is given but no information on the results. At least relative Efficiency of the detector should be presented.

Response: We, the Bangladesh Atomic Energy Commission (BAEC), are responsible for most of Bangladesh's nuclear-related activities. We have many labs throughout Bangladesh. We have many detectors, and many scientists are working in BAEC. We have to characterize all types of imported food samples with many matrices. We have many standard sources since we have to measure different matrix samples. This research study was designed, and we collected soil samples from Sundarbans. In the beginning, we used IAEA reference samples RGU-1, RGTh-1 and RGK-1 for the efficiency calibration for this particular research. Then we also cross-checked with our Eu-152 spiked source. Previously just the Eu-152 standard source was mentioned, now based on both reviewer's comments, the description of the detector efficiency determination has been modified in the revised manuscript like this (Page No. 7, Line No. 185-189), "Efficiency of the detector was determined using IAEA reference samples RGU-1, RGTh-1 and RGK-1[28]. Besides, efficiency data was also checked by a standard source which was made by combining 152Eu of known activity (Liquid form, 900 Bq activity) with the Al2O3 matrix and manufactured in the same containers as the samples."

The relative Efficiency of the detector was already added in the manuscript (Page No.6, Line No. 147). 

Reviewer1, comment 5: The number of decimal places in the results must be ordered. It is inconsistent and partially incorrect in the results. The result and the uncertainty of the result should have the same number of digits. If the result is in the hundreds, there is no need to give two decimal places; they are less than 1% of the result. General rule is that: two or three significant digits are enough.

Response: We thank the reviewer for this opportunity to clarify this issue and revised accordingly in Table 1 (Page No. 9, Line No. 239).

Reviewer1, comment 6: In the discussion of the results, it is stated many times that the obtained results exceed the limits. But the degree of this excess is not specified. The reader must analyze the whole tables with results and look for reference values in the text. You can specify the percentage range by which the results exceed the accepted limits. As it currently stands, it is not clear whether these are serious surpluses or just a few percent. This can only be guessed on the basis of the recommendations given by the authors at the end of the text.

Response: We thank the reviewer for this opportunity to clarify this issue. The degree of the obtained results exceeds the population-weighted world average values of 226Ra, 232Th and 40K has been added in the revised manuscript like this (Page No.10, Line No. 244), "The highest activity of 226Ra, 232Th and 40K obtained in the present study is 35, 51 and 660 Bqkg-1 respectively which exceeds the population-weighted world average values of 32, 45,420 for 226Ra, 232Th and 40K by 9%, 11% and 57%."

Reviewer1, comment PDF file:

Page No. , Line No. Comment Response

1, 27 and the. Space is missing.

 Corrected

2, 38 space missing

 Corrected

2, 43 According to the PLOS guideline references should be numbers in square brackets.

 Corrected throughout the manuscript

4, 107 Why capital letter?

 Corrected

 Is it number of literature reference? If yes it should be in the end of "Peat soils" or in the end of the paragraph. Or is it numbering from the following paragraph. If yes it is inconsistent. 

 Deleted 

4, 110 Are these references only for Acid Sulphate Soils? If so, then the other soils should also have appropriate references. If 17, 18 applies to all the types of soil mentioned, the reference is in the wrong place. It should be at the end of the paragraph.

 Reference has been given at the end of the paragraph.

 Suggestion: Using the way of citing as an internal numbering system may be confusing. If you change () into [] for literature references, problem will be solved.

You can also add soil type number to the previous paragraph, but keep the proper order between these two paragraphs. Now there is inconsistency.

 Corrected, description of the soil type has been deleted.

5-6, 131-134 In picture 2 there is an additional stage of drying in the sun? Why is it omitted in the description? Why were the samples additionally dried in the sun, and not only in the oven? What was the drying temperature? Some nuclides escape with vapors at too high a temperature, therefore information about the drying temperature is very important.

 Drying temperature has been included in the revised manuscript

6, 135-136 Keeping such samples one month tightly closed is generally good practice, but at least 30 days is quite imprecise. 1 year is also "at least 30 days" ;). Please, provide more detailed time period.

 Corrected, at least word has been deleted.

 The statement is very general. What do you mean, that you avoid contamination when sampling or measurements? How contamination during gamma spectrometry with HPGe measurements may be done?

 Line deleted

6, 147-152 Please provide energy lines with one equal accuracy. Accuracy down to 1 keV is fine. In practice, an accuracy of 0.01 keV is not needed to nuclide identification.

 Corrected, fraction has been deleted

 What do you mean by weighting if the secular equilibrium was achieved?

 This line has been deleted.

 Here you use the same numbering system as literature references and soil types.

 Corrected

6, 147-150 What about the background correction? Details about background and background correction has been added in the revised manuscript.

7, 163 Why in the text is MDAC? MDAC is the same as MDA?

 Corrected

 There is no "I" in both equations.

 Corrected

7, 188 Provide the activity of calibration source.

 Provided

7, 187 Please, provide a justification for this value i.e. proper reference. Properly cited

8, 197 "specific activity" should be named activity concentration of concentration of activity. Please use consistent terminology. Previously specific activity did not appear.

 Corrected

8, 202-204 Please, provide a proper reference for this equation.

 Properly cited

 If other symbols "have their usual meaning" then dose Dout should be in Bq/kg... not nGy/h. Line deleted

8, 210 If measured quantities can be used, that in your work you cannot calculate annual effective dose. You assessed Din and Dout, not measured. Corrected, used assessed

 Hin is about not only radon, in the equation itself there is potassium-40 concentration as well.

 Radon deleted

 It is inappropriate expression. Use rather radon progeny, or radon decay products, or radon daughters. Radon deleted

7, 175-178 Is it standard deviation? Or standard error of mean? Please explain.

 This is the uncertainty of the measured radioactivity which description is given in Equation 3

11, 251-252 Such a statement needs to be justified any proper reference. 

 Properly Cited 

 137Cs origin is predominantly global fallout, not only Chernobyl and Fukushima... In your work there is no information on 137Cs research, so where the statement comes from? 

 Line deleted

12, in Table 2 ? Deleted

Reviewer #2: 

Reviewer 2, comment 1:

Row 19 and rows 171-171. One cannot measure directly Th-232 via gamma radiation. What is measured are Th series progenies (Ac-228 and Tl-208, depending which lines you took under consideration). Due to high risk of lack of equilibrium in open environment they might reflect the activities of Ra-228 (case of Ac-228 lines) or Th-228 (case of Tl-206 lines). The Th-228 and Ra-228 in fresh sediment samples could not be in equilibrium, so one should present the results from Ac-228 (which could be named "Ra-228") and Tl-208 (which could be named "Th-228") separately.

Response: To the extent that you have provided feedback, I am grateful. You have been extremely helpful, and we appreciate your time and insight. Based on your suggestion, characteristic gamma lines 911 keV and 969 keV for 228Ac, were used to determine the 232Th activity concentration. As a result, Table 1, Table 2 and Table 3 data and associated lines have been revised throughout the manuscript.

Reviewer 2, comment 2: Row 28. Space between "etc." and "Elevated". The lack of space after dot or between the number and unit (for instance row. 66) repeats in many places in text, please check and correct all. BTW – there is no really elevated levels among the results.

Response: Lack of space problem has been corrected in the revised manuscript. The problem may arise because of the use of different ms word versions of the author and the reviewer. One line has been added in the Results and Discussion part like this (Page No.10, Line No. 244-247), “The highest activity of 226Ra, 232Th and 40K obtained in the present study is 35, 50 and 660 Bqkg-1 respectively which exceeds the population-weighted world average values of 32, 45, 420 for 226Ra, 232Th and 40K by 9%, 11% and 57%.”

Reviewer 2, comment 3: Row. 153. Ra-226 and Ra-224 are parents, not daughters of mentioned Rn-222 and Rn-220, respectively. You cannot call them also "elements", since the mass number is specified (nuclide, radionuclide, radioisotope or isotope are far more proper).

Response: We thank the reviewer for this opportunity to clarify this issue. The line has been modified in the revised manuscript like this (Page No.5, Line No. 135-137), “Then they were kept for 30 days to ensure that 226Ra and 232Th were in secular equilibrium with short-lived daughter products.”

Reviewer 2, comment 4: The way how the equation 1 is written could be misleading. When I have looked on denominator I've seen "exp", what is nonsense. With more careful look I realized, that "x" here is multiplication symbol. The "cps" in numerator is also not sure, if it is a product of three values or three character symbol. I suggest to re-write this formula avoiding many character symbols (instead pleas use the single character symbols with indexes) and please replace the "x" by dots. Please use the same style (italic) for symbols in all equations and in all description of them (incl. eq. 4 and 5). Please also note, that "mass" and "weight" are not the same things in physics, the proper term here is "mass". Please correct this way all equations. 

Response: We are very sorry for these silly mistakes and revised them accordingly.

Reviewer2, comment 5: Row 201. The usage of Eu-152 for efficiency calibration is not proper due to cascade transitions in decay of this isotope. One can do corrections for it using one of several methods. It looks that you have not done that correction. Without a cascade correction some of your results could by biased for even 10%.

Response: We, the Bangladesh Atomic Energy Commission (BAEC), are responsible for most of Bangladesh's nuclear-related activities. We have many labs throughout Bangladesh. We have many detectors, and many scientists are working in BAEC. We have to characterize all types of imported food samples with many matrices. We have many standard sources since we have to measure different matrix samples. This research study is designed, and we collected soil samples from Sundarbans. In the beginning, we used IAEA reference samples RGU-1, RGTh-1 and RGK-1 for the efficiency calibration for this particular research. Then we also cross-checked with our Eu-152 spiked source. Previously just the Eu-152 standard source was mentioned, now based on both reviewer's comments, the description of the detector efficiency determination has been modified in the revised manuscript like this (Page 7 , Line 185-189), "Efficiency of the detector was determined using IAEA reference samples RGU-1, RGTh-1 and RGK-1[28]. Besides, efficiency data was also checked by a standard source which was made by combining 152Eu of known activity (Liquid form, 900 Bq activity) with the Al2O3 matrix and manufactured in the same containers as the samples."

Reviewer 2, comment 5: Table 1. – the presentation of results is not proper. The data for Th series should be presented separately for Ac-228 and Tl-208 lines. 

Response: We appreciate the reviewer's feedback. Based on your suggestion, characteristic gamma lines 911 keV and 969 keV for 228Ac, were used to determine the 232Th activity concentration. Therefore, table 1, Table 2 and Table 3 data and associated lines have been revised throughout the manuscript.

Reviewer 2, comment 6:In general the results should be presented in way: xab.c±y.z or xab±vw but never xab±fg.h. The rule is: present the uncertainty only in two meaning digits. The way of presentation of main part of result is driven by the precision of uncertainty. In particular, 500±60 is OK, but 490±58.8 is not OK. The 33±3.63 is not OK as well. But 40±6 can be accepted.

Response: We appreciate the reviewer's feedback and corrected it accordingly.

---

## [Decision Letter · Decision Letter 1]

1 Feb 2023

PONE-D-22-26012R1Elevated Levels of 40K in the tourist-captivating Karamjal and Harbaria sites of the World’s largest mangrove forest SundarbansPLOS ONE

Dear Dr. Mahfuz,

Thank you for submitting your manuscript to PLOS ONE. After careful consideration, we feel that it has merit but does not fully meet PLOS ONE’s publication criteria as it currently stands. Therefore, we invite you to submit a revised version of the manuscript that addresses the points raised during the review process.

We look forward to receiving your revised manuscript.

Kind regards,

Mohamad Syazwan Mohd Sanusi

Academic Editor

PLOS ONE

Journal Requirements:

Additional Editor Comments:

Dear author, you are invited to revise your manuscript as per commented by the reviewers

Reviewers' comments:

Reviewer's Responses to Questions

**Comments to the Author**

1. If the authors have adequately addressed your comments raised in a previous round of review and you feel that this manuscript is now acceptable for publication, you may indicate that here to bypass the “Comments to the Author” section, enter your conflict of interest statement in the “Confidential to Editor” section, and submit your "Accept" recommendation.

Reviewer #1: (No Response)

Reviewer #2: All comments have been addressed

2. Is the manuscript technically sound, and do the data support the conclusions?

Reviewer #1: Yes

Reviewer #2: Yes

3. Has the statistical analysis been performed appropriately and rigorously? 

Reviewer #1: N/A

Reviewer #2: Yes

4. Have the authors made all data underlying the findings in their manuscript fully available?

Reviewer #1: Yes

Reviewer #2: Yes

5. Is the manuscript presented in an intelligible fashion and written in standard English?

Reviewer #1: Yes

Reviewer #2: (No Response)

6. Review Comments to the Author

Reviewer #1: Dear Authors,

I was pleased to read your work once again. I fully sustained my opinion about the scientific value of your work, the way of presenting it in the manuscript etc. You have exhaustively addressed my previous comments and corrected and completed the manuscript. I just have only a few more minor remarks:

p. 7 line 167 – in the equation (2) in the denominator there are simple dots, not a multiplication symbol

p. 7 line 175 - In formula (3) there is a collision of symbols. N in formula (1) is net count rate, here ( in 3)it is sample count. If N in formula (1) and (3) means the same, then formula (3) implicitly doubles the time factor. The first time implicitly is in u(N)/N and the second time explicitly u(T)/T. This falsifies the uncertainty value. If N in formula (1) stands for net counts per time and in (3) gross counts (area under the peak in the spectrum), then formula (3) is correct but it is necessary to change the notation to eliminate confusion. Please clarify this issue.

p. 10 Table 1 – you present here Average±?.You have stated that number after “±” “This is the uncertainty of the measured radioactivity which description is given in Equation 3”. If it is the uncertainty of the measured radioactivity, how did you measure the average activity? I can guess how to calculate it, but measure? And how did you determine all the factors in the formula since they relate to the average? Measure of uncertainty of average it is usually standard deviation (if normal distribution is dealt with) or standard error (standard error of mean/standard deviation on mean).

p. 13 line 292 – reference [62] does not appear in the table 2 and is not about Rio Formoso but coastal Karnataka.

p. 17 line 368 – “no radioactivity of 137Cs was detected in the collected soil samples” We only learn about it from the Conclusion. This information should be included in the Results. Especially since you state in the Introduction that: “Therefore, this present study aims […] to check if any artificially occurring 137Cs is present.” You can also provide average MDA for the 137Cs in your measurements, after all we cannot know if there is 137Cs or not, we only know the level of 137Cs was below the MDA.

At the end the suggestion I have missed previously. If you name in the title 40K, and it is the key point result in your work consider adding 40K to the key words of your article.

Reviewer #2: Dear Authors. Thank you for your answers, which seems to clarified all my doubt. I think that corrected paper can be accepted for publication in PLOS.

7. PLOS authors have the option to publish the peer review history of their article (what does this mean?). If published, this will include your full peer review and any attached files.

Reviewer #1: No

Reviewer #2: No

---

## [Author Response · Author response to Decision Letter 1]

4 Feb 2023

Response to reviewer(s) 

Respected Reviewers,

Thank you so much for your comments. We are very grateful to you for the time and intelligence that you have shared with us. It is a great learning opportunity for us through these reviews. Our responses to the comments are highlighted in red color in the revised manuscript.

Comments and Suggestions for Authors:

Reviewer #1 Comment:

Dear Authors, 

I was pleased to read your work once again. I fully sustained my opinion about the scientific value of your work, the way of presenting it in the manuscript etc. You have exhaustively addressed my previous comments and corrected and completed the manuscript. I just have only a few more minor remarks:

Response: We are grateful to the reviewer for this valuable comment.

Reviewer 1, comment 1: p. 7 line 167 – in the equation (2) in the denominator there are simple dots, not a multiplication symbol 

Response: We thank the reviewer for indicating this issue. In the revised manuscript, we have changed the dots into multiplication symbol of the denominator in the both Eq (1) (Page No.6, Line No. 160) and Eq (2). (Page No.7, Line No. 167). 

Reviewer 1, comment 2: p. 7 line 175 - In formula (3) there is a collision of symbols. N in formula (1) is net count rate, here (in 3) it is sample count. If N in formula (1) and (3) means the same, then formula (3) implicitly doubles the time factor. The first time implicitly is in u(N)/N and the second time explicitly u(T)/T. This falsifies the uncertainty value. If N in formula (1) stands for net counts per time and in (3) gross counts (area under the peak in the spectrum), then formula (3) is correct but it is necessary to change the notation to eliminate confusion. Please clarify this issue.

Response: We appreciate the opportunity to explain this matter to the reviewer. We have changed the net counts per sec in equation (1) from ‘N’ to ‘Z’ to avoid confusion. (Page No.6, Line No. 160-163)

Reviewer 1, comment 3: p. 10 Table 1 – you present here Average±?. You have stated that number after “±” “This is the uncertainty of the measured radioactivity which description is given in Equation 3”. If it is the uncertainty of the measured radioactivity, how did you measure the average activity? I can guess how to calculate it, but measure? And how did you determine all the factors in the formula since they relate to the average? Measure of uncertainty of average it is usually standard deviation (if normal distribution is dealt with) or standard error (standard error of mean/standard deviation on mean).

Response: We thank the reviewer for this opportunity to clarify this issue. In order to calculate the average activity; and the average uncertainty of the measured radioactivity, we simply calculated the mean. 

Reviewer 1, comment 4: p. 13 line 292 – reference [62] does not appear in the table 2 and is not about Rio Formoso but coastal Karnataka.

Response: We thank the reviewer for this valuable comment and revised accordingly (Page No. 13, Line No. 294-295).

Reviewer 1, comment 5: p. 17 line 368 – “no radioactivity of 137Cs was detected in the collected soil samples” We only learn about it from the Conclusion. This information should be included in the Results. Especially since you state in the Introduction that: “Therefore, this present study aims […] to check if any artificially occurring 137Cs is present.” You can also provide average MDA for the 137Cs in your measurements, after all we cannot know if there is 137Cs or not, we only know the level of 137Cs was below the MDA. At the end the suggestion I have missed previously. If you name in the title 40K, and it is the key point result in your work consider adding 40K to the key words of your article.

Response: We appreciate the reviewer for allowing us to clarify this issue. In the revised manuscript, we have replaced the statement in the introduction (Page No. 4, Line No. 93): “… … to check if any artificial radionuclide is present… …” and in the conclusion (Page No. 17, Line No. 370-371): “In regard to anthropogenic activity, no artificial radionuclide was detected in the collected soil samples.” We have added the following lines in the result and discussion of the revised manuscript (Page No. 11, Line No. 267-268): “No artificial radionuclides were detected in the collected samples from the Sundarbans.”

We have also added 40K to the keywords in the revised manuscript (Page No. 2, Line No. 33)

Reviewer #2 Comment: 

Dear Authors. Thank you for your answers, which seems to clarified all my doubt. I think that corrected paper can be accepted for publication in PLOS.

Response: We are grateful to the reviewer for this valuable comment.

---

## [Editor Report · Decision Letter 2]

20 Mar 2023

PONE-D-22-26012R2Elevated Levels of 40K in the tourist-captivating Karamjal and Harbaria sites of the World’s largest mangrove forest SundarbansPLOS ONE

Dear Dr. Mahfuz,

Thank you for submitting your manuscript to PLOS ONE. After careful consideration, we feel that it has merit but does not fully meet PLOS ONE’s publication criteria as it currently stands. Therefore, we invite you to submit a revised version of the manuscript that addresses the points raised during the review process.

We look forward to receiving your revised manuscript.

Kind regards,

Mohamad Syazwan Mohd Sanusi

Academic Editor

PLOS ONE

Journal Requirements:

Additional Editor Comments:

Kindly find my full comment for minor revision.
---

## [Author Response · Author response to Decision Letter 2]

20 Apr 2023

Response to Editor's Comments

Respected Editor,

We greatly appreciate your time that spent on our manuscript and sharing your knowledge with us to further improve our manuscript. We have an excellent opportunity to learn from these reviews. In the modified manuscript, our responses to the comments are indicated in red color. 

Comments and Suggestions for Authors

Editor comment 1: Please revise the title. The title must reflect the study contents.

1) Measurement of K-40 activity in Karamjal and Harbaria mangrove forest, Sundarbans to investigate the downhill agriculture activities impacts.

Or,

2) Measurement of soil radioactivity in Karamjal and Harbaria mangrove forest, Sundarbans for radiological database establishment

Response: Thank you so much for your kind comments and appreciation of our study. All of your comments are carefully evaluated and revised in our revision accordingly. The title has been revised based on your 2nd recommendation, “Measurement of radioactivity in soils of Karamjal and Harbaria mangrove forest of Sundarbans for establishment of radiological database”

Editor comment 2: The abstract lacks of problem statement and objective, methodology, results, limited discussion.

Response: We thank the editor for this comment. The abstract has been revised according to the guideline of the Editor, “This work presents the first in-depth study of soil radioactivity in the mangrove forest of Bangladesh part of the Sundarbans. It used HPGe gamma-ray spectrometry to measure the amount of natural radioactivity in soil samples from Karamjal and Harbaria sites of the world's largest mangrove forest. The activity concentrations of most of the 226Ra (14±2 Bqkg-1 to 35±4 Bqkg-1) and 232Th (30±5 Bqkg-1 to 50±9 Bqkg-1) lie within the world average values, but the 40K concentration (370± 44 Bqkg-1 to 660±72 Bqkg-1) was found to have exceeded the world average value. The evaluation of radiological hazard parameters revealed that the outdoor absorbed dose rate (maximum 73.25 nGyh-1) and outdoor annual effective dose (maximum 0.09 mSvy-1) for most samples exceeded the corresponding world average values. The elevated concentration of 40K is mainly due to the salinity intrusion, usage of fertilizers and agricultural runoff, and migration of waste effluents along the riverbanks. Being the pioneering comprehensive research on the Bangladesh side of the Sundarbans, this study forms a baseline radioactivity for the Sundarbans before the commissioning of the Rooppur Nuclear Power Plant in Bangladesh.”

Editor comment 3: Do not combine the lines. Please use the active sentence, active voice constructions are usually stronger, clearer, more direct, and often more concise than their passive-voice counterparts like you did.

Response: We appreciate the editor's feedback and changed it accordingly.

Editor comment 4: This should be in the last line before conclusion.

Response: We thank the editor for this opportunity to clarify this issue. This line has been replaced before conclusion.

Editor comment 5: To state such slightly high K-40 is insufficient. To discuss in terms of increased risk of radiation exposure, you need to highlight ICRP 103 recommendations on Existing Exposure Situations and dose levels beyond 100 mSv @ mGy. Please remove the biota, since the author didn’t estimate the biota risk using ICRP and IAEA dose estimator.

Response: We thank the Editor for this opportunity to clarify this issue and respect his comment, so we have deleted the risk issue and revised our abstract. The word “the biota” has been removed.

Editor comment 6: The choice of keywords need to be identified based on potential reader interest.

Response: We appreciate the editors's feedback. We have revised our keywords like this (Page-2, Line 31-32), “Keywords: Mangrove Forest, The Sundarbans, Natural radioactivity, Gamma Spectrometry, Radiological indices, Effective Dose”

Editor comment 7: Please revise. Background radiation pose significant health risk? 

Response: We appreciate the editors's feedback and corrected accordingly (Page 2, line 38-39), “Increased use of ionising radiation or anthropogenic radioactivity may pose non-negligible threats to living beings”.

Editor comment 8: Please extend the discussion with results and finding

Response: We appreciate the opportunity to explain this matter with the editor. The discussion has been extended like this (Page 3, Line 61-67), “In the Indian region of the Sundarbans, the average activity of 40K (532 – 1043 Bqkg-1) was reported to be more than twice as high as the global average of 420 Bqkg-1 [5]. The authors hypothesised that the accumulation of upstream wastes and undesirable effluents along this coastal zone, together with rising salinity and the usage of fertilisers to boost crop productivity, may have collectively inficted K content. As the implications are comparable, it is necessary to do a similar radioactivity measurement in the Bangladesh part of the Sundarbans”. 

Editor comment 9: The exposure is insignificant due to tourist. The main problem statement to conduct this study is :

1) establish baseline data of environmental radioactivity due to newly proposed Rooppur Nuclear Power Plant in the area.

2) To assess the human activities impact (coal power plant, agriculture, uncontrolled fishimng activities) on the Sundarbands ecossyem by investigate the radioactivity distribution.

Response: We appreciate the editor's feedback. We have deleted the line, “The Sundarbans, being a World Heritage Site, attracts thousands of local and foreign tourists every year [7]. This also raises concerns that the radioactivity that the tourist is exposed to is unknown” and revised the manuscript like this (Page 4-5, line 107-112): “Therefore, this present study aims to measure the prevailing concentration of NORMs in the soils of Sundarbans, the first of its kind, to assess the impact of human activities (coal fired power plant, agriculture, uncontrolled fishing activities) on the Sundarbands ecosystem by investigating the radioactivity distribution. This study also aims to provide baseline data which is important due to the recent commissioning of a nuclear power plant and several thermal power plants surrounding the Sundarbans.”

Editor comment 10: Please report the results and finding from these countries

Response: We respect the Editor's comments. The results and findings from these countries has been added in the revised manuscript like this (Page 3, line 83-105), “A few studies were reported on radiation levels of the mangrove forests in Pernambuco, Brazil, where maximum level of 40K was found to be 1338 Bq kg-1, and the authors concluded that this was due the influence of sediments and presence of granites [8]. Radioactivity studies conducted in the petrified wood forests in Egypt found levels of 238U (65.26±12.99 Bqkg-1) exceeding the world average values, along with the presence of the artificial radionuclide 137Cs arising from the nuclear accident in Chernobyl, and from the deposition from nuclear weapon tests in the neighbouring countries [9]. Very high concentrations of 232Th were found in three Norwegian forests due to their proximity to an active volcano and the complex mixture of heavy mineral salts present there [10]. In the Indian Sundarbans mangrove forest, large amounts of 40K were discovered (maximum 866 Bqkg-1), mostly due to the continuous deposition and erosion of silt, sediment, and other organic matter [1], also low levels of NORMs were discovered in the Krusadai Island Mangrove as a result of the region's low concentration of radiation-bearing minerals [11], and nitrogen fertilisers, a source of 40K in the environment, are widely used to improve soil nitrogen balance for agricultural development [12]. Moreover, it was found that the activity of 238U and 232Th had decreased from pre-tsunami data (25.9± 15.8 Bqkg-1 and 65.1±34.5 Bqkg-1) to post-tsunami data (12.2± 4.2 Bqkg-1 and 11.7±5.0 Bqkg-1 [13]. Although, the conditions are similar in the Sundarbans of Bangladesh only two studies have been found to date in the Sundarbans of Bangladesh site, one of which reported levels of NORMs were below the world average values in sediment samples [14] and the other study assessed the number of trace elements in sediments samples revealing, using various environmental contamination indices, that the sediments of the region have moderate to severely contaminated levels of Cd and moderate to low levels of As, Sb, Th, and U [15].

Editor comment 12: Please clarify does this impact the activity levels of the places?

Response: We respect the Editor's comments and deleted the line.

Editor comment 13: to assess the human activities impact (coal power plant, agriculture, uncontrolled fishimng activities) on the Sundarbands ecosystem by investigate the radioactivity distribution

Response: We have revised the line according to the Editor’s guideline (Page 4-5, Line 107-112), “Therefore, this present study aims to measure the prevailing concentration of NORMs in the soils of Sundarbans, the first of its kind, to assess the impact of human activities (coal fired power plant, agriculture, uncontrolled fishing activities) on the Sundarbands ecosystem by investigating the radioactivity distribution. This study also aims to provide baseline data which is important due to the recent commissioning of a nuclear power plant and several thermal power plants surrounding the Sundarbans."

Editor comment 14: Don’t overclaim the title of this section, the discussion is general..

Response: We thank the editor for this comment. We have deleted the term “Geomorphology, pedology” in the revised manuscript. 

Editor comment 15: Remove, merge 2.1 and 2.2 Not related. Th provided map is a topography information. 

Response: We thank the editor for this comment. The heading “2.2 Geomorphological map preparation” has been removed and we have merged 2.1 and 2.2. Other numbering of subsections has also been changed accordingly.

Editor comment 16: How do you justify 30 samples? Why? Statistical validation or for normality test? How do you determine on these location, as the Sundarbans area is 10,000 square km, why 2 distinctive locations ? The study aim to investigate the variations across uphill and downhill?

Response: We thank the editor for this comment. For this particular research, we have chosen two tourists spot located at Karamjal and Herbaria in Sundarbans. Because of the safety purpose, the designated area for visitors for these particular tourists’ spots is very limited, so we have collected 15 samples from each of the tourist spot. 

Editor comment 17: Criteria of samples? Top soil? Figure 1 shows sampling at Karamjal is situated on the river and riverbank? As the study aim to study the agriculture human impacts etc., thus top soil sampling should be consider.

Response: We thank the editor for this comment. Yes, it is topsoil; so, the quoted sentence has been rephrased as follows (Page 6, Line 136): " A total of 30 topsoil samples were collected from the area around the Sundarbans (15 from Karamjal and 15 from Harbaria) in December 2021,"

Editor comment 18: Systematic or gridded sampling is different to random. Please clarify and revise

Response: We thank the editor for this comment. In IAEA Technical Reports Series No. 486, “Guidelines on Soil and Vegetation Sampling for Radiological Monitoring”, the following sampling methods are described: Simple random sampling; Two-stage sampling; Stratified sampling; Systematic grid sampling; Systematic random sampling; Cluster sampling; Double sampling; Search sampling and Transect sampling. Following the IAEA guideline, we adopted systematic random sampling in this research. According to this IAEA guideline, systematic random sampling is a useful approach in estimating the average concentration within grid cells. The area is divided using a square or triangular grid, and then samples are collected at random locations from within each cell using the same procedures as simple random sampling. This method is useful in delineating the extent of the contamination, confirming cleanup and for field screening. Hope this clarify the raised issue.

Editor comment 19: Explain the line, mixed with what material? 

Response: We thank the editor for this comment. Actually, it’s a mistake. No extra thing was mixed, just made the sample homogeneous, with the help of sieve. It should be homogenously mixed, so we revised the line (page 6, line 140-141), “After removing extraneous components like roots, pebbles, and plant matter, along with other impurities, the samples were homogenously mixed.”

Editor comment 20: Marinelli beaker?

Response: We thank the editor for this comment. Yes, it is Marinelli beaker and the line has been revised like this (Page 6, line 146-147): “All samples were then put into radon-impermeable, airtight Marinelli beakers (EG &G, Ortec)”. 

Editor comment 21: Use Bq kg-1 consistent with activity in results

Response: We appreciate the editor's feedback and corrected the whole manuscript using 

Bqkg-1. 

Editor comment 22: Is this in equation 1? How do you obtain the value of the detector efficiency for each of gamma energy line?

Response: We appreciate the editor's comment. Yes, this is ε in Equation 1 which is the detectors counting efficiency. The efficiency of the detector for each of gamma energy line was obtained from the efficiency vs energy curve using IAEA standards (RGU-1, RGTh-1, and RGK-1.)

Editor comment 23: How the high energy gammas are calibrated? The calibration process does not address the instrument's performance at higher energies, namely, K-40 1460keV, Bi-214 1764keV and Tl-208 2614keV. These energies contribute significantly to air kerma rates from NORM.

Response: We thank the editor for this comment. Besides these common point sources 22Na, 57Co, 60Co, 133Ba, 137Cs, we also use 152Eu and 88Y for high energy gamma ray calibration. We have added these point sources in the revised manuscript (Page no. 8, Line no. 195-197): “The detector’s energy calibration was performed using common point sources like 22Na, 57Co, 60Co, 88Y, 133Ba, 137Cs, 152Eu, etc.” 

Editor comment 24: How the efficiency calibration was conducted? Give details procedure that benefit the readers. And why need to proceed with secondary validation using standard Eu-152? Generally, only one standard sample need to be used as correction reference for the uncalibrated counting sample

Response: We appreciate the editors's input. The quoted sentence has been rephrased as follows (Page 8, Line 199-202): “The IAEA standards (RGU-1, RGTh-1, and RGK-1) were used to determine the counting efficiency, which are Certified Reference Materials (CRM) and contain natural radionuclides from 238U-series, 232Th-series, and 40K, whose certified activity concentrations were 4940 ± 15 Bqkg-1, 3250 ± 45 Bqkg-1,14000 ± 200 Bqkg-1.”

We, the Bangladesh Atomic Energy Commission (BAEC), are responsible for most of Bangladesh's nuclear-related activities. We have many labs throughout Bangladesh. We have many detectors, and many scientists are working in BAEC. We have to characterize all types of imported food samples with many matrices. We have many standard sources since we have to measure different matrix samples. This research study was designed, and we collected soil samples from Sundarbans. In the beginning, we used IAEA reference samples RGU-1, RGTh-1 and RGK-1 for the efficiency calibration for this particular research. Then we also cross-checked with our Eu-152 spiked source (152Eu of known activity (Liquid form, 900 Bq activity) with the Al2O3 matrix and manufactured in the same containers as the samples). 

Editor comment 25: Please include aim of this? For interpolation on unsampling location or just a representation of small scale sampling. Include the IDW parameters: Power? Neighbors search number? Direction? Angle? 0? Or isotropic

Response: We appreciate the editor's input. Subsection 2.7 has been revised like this (Page 10, Line-245-250), “For interpolation of some derived data like radium equivalent activity, absorbed dose rate and annual effective dose in unsampling locations within the study area, interpolation was carried out using ArcGIS 10.2 software. The inverse distance weighting (IDW) technique was applied to interpolate the value of a variable at unmeasured sites from observations of its values at nearby locations according to our previous study [35], [36].”

Editor comment 26: Any details explanation on this? Solubility with water or simple ionic exchange process with h20?

Response: We thank the editor for this comment. 226Ra readily dissolves in groundwater so that 226Ra may experience surface runoff on muddy terrain.

Editor comment 27: Should be in Line 246

Response: Corrected (Page 11, line 259-262)

Editor comment 28: All the maps required major correction. The only reference is longitude and latitude, please consider the boundary line, river, soil, sampling point, as in Fig 1

Response: We thank the editor for this comment. All the maps have been corrected based on your suggestion.

Editor comment 29: What is the average value reported by UNSCEAR. The effective dose is small and less than 1 mSv, how lot of time they spend outdoor? 12 hours a day? The dose seems at acceptable range. To discuss in term of risk, I suggest the author to use the ICRP discussion on Existing Exposure Situation of 1-20 mSvy-1

Response: We respect the editors’s feedback regarding the issue and we have decided to delete the effective dose issue and revised the manuscript like this (Page 18, line 387-388) “Most of the hazard parameters are within the recommended safety limits, so the research areas are not posing a considerable hazard for short-term stay by visitors.”

---

## [Editor Report · Decision Letter 3]

12 Jul 2023

Measurement of radioactivity in soils of Karamjal and Harbaria mangrove forest of Sundarbans for establishment of radiological database

PONE-D-22-26012R3

Dear Dr. Siraz,

We’re pleased to inform you that your manuscript has been judged scientifically suitable for publication and will be formally accepted for publication once it meets all outstanding technical requirements.

Kind regards,

Md. Naimur Rahman

Academic Editor

PLOS ONE

Additional Editor Comments (optional):

The revision version of the manuscript was satisfactory. The manuscript is accepted. Thank you for your submission.
---

## [Editor Report · Acceptance letter]

14 Jul 2023

PONE-D-22-26012R3 

Measurement of radioactivity in soils of Karamjal and Harbaria mangrove forest of Sundarbans for establishment of radiological database 

Dear Dr. Siraz:

I'm pleased to inform you that your manuscript has been deemed suitable for publication in PLOS ONE. Congratulations! Your manuscript is now with our production department. 

Kind regards, 

on behalf of

Mr Md. Naimur Rahman 

Academic Editor

PLOS ONE